# Formal Concept Lattices are Good Semantic Scaffolds for Concept-Based Learning

Deepika SN Vemuri [1]   Sayanta Adhikari[† 1 2]   Ankit Saha[†‡ 1]   Krishn Vishwas Kher [1]
Vineeth N Balasubramanian [1 3]

## Abstract

Learning semantics is essential for deep learning models to be interpretable and better aligned with human reasoning. Concept-based models approach this by representing classes through meaningful semantic abstractions, but typically treat all concepts as a flat, unstructured set learned at a single neural network layer. This overlooks a fundamental property of human semantic understanding: concepts being organized hierarchically, from general to specific. While deep networks do learn a hierarchy of visual features, this structure is rarely aligned with explicit semantic hierarchies. Drawing on Formal Concept Analysis, we demonstrate that formal concept lattices provide principled semantic scaffolds to guide neural network learning. These lattices naturally identify where in the network concepts should be learned based on their level of generality. This allows the model to develop staged, semantically grounded representations throughout its depth. Empirical results on real-world datasets show that our models produce more interpretable embeddings, support more effective interventions, and learn concept representations that are both meaningful and hierarchically structured. Code available at: https://github.com/deepikavemuri/FoCA-CBMs.

## 1. Introduction

For many years now, deep neural networks (DNNs) have been known to learn hierarchical representations. In computer vision, the early layers of these networks capture generic features like texture, while the later layers encode class-specific information (Zeiler & Fergus, 2014; Olah et al., 2018). However, the exact nature of these representations remains opaque and semantically less interpretable.

[†] Work started while the authors were students at IITH.
[‡] Currently works at KLA. [1]IIT Hyderabad [2]Amazon, India [3]Microsoft Research. Correspondence to: Deepika SN Vemuri <ai22resch11001@iith.ac.in>, Vineeth N Balasubramanian <vineethnb@cse.iith.ac.in, vineeth.nb@microsoft.com>.

*Proceedings of the 43rd International Conference on Machine Learning*, Seoul, South Korea. PMLR 306, 2026. Copyright 2026 by the author(s).

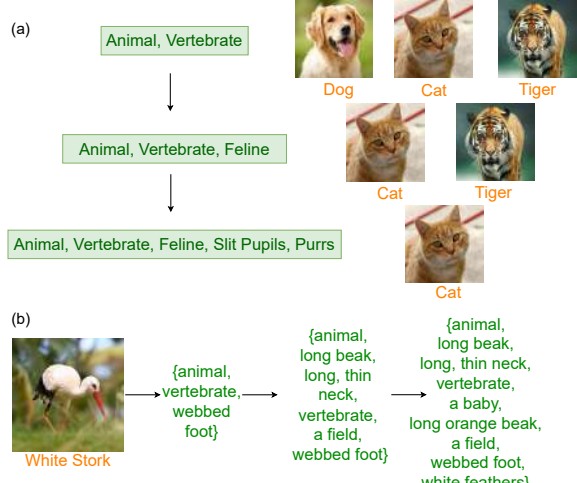

*Figure 1.* Classes (orange) and attributes (green) (a) An illustrative example of how attributes shared by more classes are more general; those shared by fewer are more specific, naturally forming a subset-superset hierarchy; (b) Attributes learned by a FoCA CBM at different layers for the class *White Stork* in ImageNet100.

Recent work has focused on making models inherently interpretable (Chen et al., 2019; Sarkar et al., 2022) so as to have better insight into what these models are learning. In particular, concept-based models or CBMs (Koh et al., 2020; Oikarinen et al., 2023; Liu et al., 2025) have emerged as a promising direction that quantify how learned concepts contribute to predictions. However, existing concept-based models typically learn all concepts at a single layer, overlooking the inherent hierarchical structure in neural network representations across multiple layers.

Human cognition, on the other hand, organizes knowledge through semantic hierarchies and reasons over different levels of learned concepts (Theves et al., 2021) (see Fig 1a). Existing CBMs do not leverage such structure; while a few limited recent efforts have explored hierarchical concept sets (Panousis et al., 2024; Sun et al., 2024), architecturally, all concepts are still learned at the same stage in the network - immediately before classification. In contrast, in this work, we examine *how concept learning can capture hierarchical structure across network depth*. To this end, we explicitly guide the network to learn general human-understandable

concepts in early layers and specific ones in deeper layers and see that this helps the model learn more semantically grounded representations while additionally allowing interpretability at different granularities.

In classification tasks where each class is defined by a set of attributes, a natural semantic hierarchy emerges from the pattern of attribute sharing. Attributes shared by many classes are general, while those shared by a few are specific. For instance, a *cat* and *tiger* are an *animal, vertebrate* and are *feline*; while a *dog*, *cat* and *tiger* are an *animal* and are *vertebrate*. Here, the latter group {*animal, vertebrate*} is more general as it spans more classes. Conversely, more specific attributes help better class discriminability in data samples. This subset-superset structure creates a concept hierarchy where generality corresponds to the number of classes sharing an attribute set, as shown in Fig 1.

To leverage such semantic structure while learning DNN models, we draw on Formal Concept Analysis (FCA) (Ganter & Wille, 2024) to construct a concept lattice from class-attribute associations. The lattice identifies natural supervision points in the network by aligning with class density patterns across network depth. Supervisory signals are then extracted from the lattice to learn sets of attributes and classes at these supervision points. These sets define *hierarchical semantic layers*, each comprising an attribute layer and a classifier layer, that effectively overlay a semantic scaffold onto the network's visual feature hierarchy. (Fig 1b illustrates the progressively refined attribute sets for the class *White Stork*). Each attribute layer corresponds to (and hence predicts) a group of classes, with the group's specificity determined by the granularity of its attributes. This layered structure enables progressive refinement of class predictions, while improving model transparency. One could view our formulation as generalizing extant CBM paradigms by exploiting a dataset's concept taxonomy. When the taxonomy is flat, it defaults to the canonical single-layer concept representation characteristic of traditional CBMs.

**Our Contributions:**

- We generalize the notion of concept-based interpretability in DNNs to semantic hierarchies using concept lattices, thus providing a means to leverage semantic scaffolds to guide learning across layers and enabling a deeper notion of interpretability in such models.
- We theoretically analyze our approach to study why such semantic ordering matters for concept-based models.
- Through comprehensive experiments on benchmark datasets, we show that this approach to learning not only performs on accuracy, but also yields semantically more meaningful embeddings, which we show using a clustering analysis.
- As part of a range of ablation studies and analysis, we show that our framework provides a mechanism to conduct multi-level concept interventions, going beyond the standard single-layer interventions in existing CBMs.

## 2. Related Work

**Concept-Based Models.** Building inherently interpretable models using concepts is an actively growing area of research, initially introduced as the idea of learning classes through a concept layer in the network (Koh et al., 2020). Follow-up works improve various aspects of these models, such as addressing concept leakage (Marconato et al., 2022), including uncertainty quantification (Kim et al., 2023) and improving robustness (Sinha et al., 2022). Other efforts include increasing model capacity using additional unsupervised concepts (Sawada & Nakamura, 2022) and building concept bases for such models (Yuksekgonul et al., 2023). More recent efforts have attempted the use of LLMs and VLMs for concept guidance and annotations (Oikarinen et al., 2023; Yang et al., 2023; Srivastava et al., 2024). Finally, some works have studied concept relations (Vandenhirtz et al., 2024b; Raman et al., 2024) and incorporating structure over concepts (Barbiero et al., 2024; De Felice et al., 2026). All these efforts focus on learning concepts at the last layer with *no semantic scaffolding across layers*. This is an aspect we focus on in this work.

**Hierarchical Learning.** Hierarchical learning has been explored more generally from a few perspectives. One line of work learns hierarchical embeddings like order embeddings (Vendrov et al., 2015), hyperbolic entailment cones (Ganea et al., 2018) and Poincaré embeddings (Nickel & Kiela, 2017). These methods, however, typically impose geometries *across samples* to align with pre-existing structure (often hierarchical) in the label space. Our approach, on the other hand, induces a concept hierarchy across features of *single sample*. Another line of work uses a hierarchy to constrain the predictions of the model (Giunchiglia & Lukasiewicz, 2020; 2022; Li et al., 2023). Some CBM-based works use hierarchical concept sets, although they are limited to two-level ones (Sun et al., 2024; Panousis et al., 2024) and are limited to the pre-classification step. In contrast, we focus on deriving supervisory signals from a structured formal concept lattice with an arbitrary number of levels (26 levels on one of the datasets we use).

**Formal Concept Analysis (FCA).** This is a mathematical theory of data analysis where a set of objects and attributes are used to derive structured hierarchy of formal concepts. There have been a few sparse efforts to use this theory in deep learning settings: to encode closure operators in a neural network (Rudolph, 2007), to introduce an embedding technique (Durrschnabel et al., 2019) for problems with formal context-like structures like bipartite graphs (Peng et al., 2024), and to obtain order-based representations using binary vectors (Gyurek et al., 2024). To the best of our

knowledge, ours is the first effort to apply ideas from FCA to a concept-based learning setting in vision to overlay a structured organization of concepts on a neural network's representations, which provides a strong semantic interpretation including at intermediate levels of a network.

# 3. Lattices for Concept-Based Learning

## 3.1. Background and Preliminaries

**Concept-Based Models:** We follow the standard CBM setup introduced by (Koh et al., 2020) and define a concept-based model as one that learns a mapping from $X \mapsto Y$ via an intermediate concept encoder $q(\cdot)$. Such models learn from a three-tuple dataset $\mathcal{D} = \{X, C, Y\}$ where $X \in \mathbb{R}^m$, $C \in \mathbb{R}^k$, $Y \in \mathbb{R}^n$ and $m, k, n$ are dimensions of the image, concept and label spaces respectively. Each prediction is of the form $\hat{y} = p(q(x))$ where $q \colon X \mapsto C$ (e.g. *bird image* $\rightarrow$ {*white body, flat yellow bill, . . . , orange legs*}) is the concept encoder, and $p \colon C \mapsto Y$ (e.g. {*white body, flat yellow bill, . . . , orange legs*} $\rightarrow$ *Duck*) is an interpretable classifier network. These text-based concepts correspond to attributes (as discussed in earlier sections), and we refer to them as such henceforth.

**Formal Concept Analysis (FCA):** A *formal context* is defined as a three-tuple $\langle G, M, I \rangle$, where $G$ is a set of objects, $M$ is a set of attributes, and $I \subseteq G \times M$ captures the binary relations (also called incidence relation) indicating which attributes are present in which objects. Given such a formal context, a *formal concept* (Ganter & Wille, 2024) is defined as a tuple $\langle A, B \rangle$, where $A$ (*extent*) is a subset of objects and $B$ (*intent*) is a subset of attributes. Note that these are not arbitrary subsets; these subsets of objects and attributes have concept-forming operators defined over them $(\uparrow, \downarrow)$. $A$ contains objects (classes in our case) sharing all attributes in $B$, and $B$ contains attributes shared by all objects in $A$. Given $A \subseteq G, B \subseteq M$, this is defined as:

$$A^{\uparrow} = B, B^{\downarrow} = A \qquad (1)$$
$$A^{\uparrow} = \{m \in M \mid \forall g \in A : \langle g, m \rangle \in I\}$$
$$B^{\downarrow} = \{g \in G \mid \forall m \in B : \langle g, m \rangle \in I\}$$

The set of all formal concepts derived from a formal context forms a partial order over the subset-superset ordering relation, i.e. if $\langle A_1, B_1 \rangle$, $\langle A_2, B_2 \rangle$ are two formal concepts, $\langle A_1, B_1 \rangle \preceq \langle A_2, B_2 \rangle$ if $A_1 \subseteq A_2$ and $B_1 \supseteq B_2$, where $\preceq$ represents subconcept-superconcept ordering. This implies that general concepts have lesser attributes and more objects, while specific concepts have more attributes and lesser objects. For example, $\langle \{dog, cat, tiger\}, \{animal, vertebrate\} \rangle$ is more general than $\langle \{cat, tiger\}, \{animal, vertebrate, feline\} \rangle$. This partial order allows us to construct a lattice of formal concepts, a hierarchy wherein concepts in higher layers are more general and ones in lower layers are more specific.

**Definition 3.1** (Formal Concept Lattice). Let $\mathcal{B}(G, M, I)$ denote the collection of all formal concepts of the formal context $\langle G, M, I \rangle$, i.e. $\mathcal{B}(G, M, I) = \{\langle A, B \rangle \in 2^G \times 2^M \mid A^{\uparrow} = B, B^{\downarrow} = A\}$. $\langle \mathcal{B}(G, M, I), \preceq \rangle$ is then a formal concept lattice (or simply lattice, in this work), where $\preceq$ is the subset-superset ordering. Let $\mathcal{L}$ denote this lattice, where $\mathcal{L} = \{\mathcal{L}_1, \mathcal{L}_2, \ldots, \mathcal{L}_L\}$. $\mathcal{L}_i$ represents the set of formal concepts at level $i$, with $i = 1$ being the most general (top) level and $i = L$ the most specific.

## 3.2. Formal Concepts for Deep Neural Networks

FCA (Ganter & Wille, 2024) provides a principled framework for organizing objects and attributes into a hierarchical structure based on their binary incidence relation. This naturally translates to a concept-based setting in DNNs where we have access to two interpretable sets: classes and their attributes. Treating classes as objects ($G$) and attributes as their properties ($M$), we define a formal context $\langle G, M, I \rangle$, where $I$ is a binary relation indicating which classes contain which attributes. From this formal context, we construct a formal concept lattice: structured tuples of class and attribute subsets organized hierarchically (see Appendix for examples of formal concepts).

Fig 2 (top, in white box) shows an illustrative example of such a lattice. The formal concepts shown at the bottom of the lattice contain singleton classes and the set of attributes that they contain. As we go up the lattice, the formal concepts become more general with increasing class set sizes along with the corresponding maximal set of attributes they have in common. Attributes individually are not general or specific; it is the set of attributes that capture levels of generality. Note that the hierarchy herein captures a subset-superset ordering: attribute sets shared by more classes are more general, while those shared by fewer are more specific. We construct such a lattice for each dataset in our experiments; more lattice construction details are provided in Appendix A6.

We next discuss how sets derived from this lattice can be mapped to specific depths in the network using a mechanism we propose called *class-cluster density*. Then, we detail our training process which involves the joint optimization of attribute learning, iterative class-group refinement and final classification.

**Extracting Attribute and Class Sets.** The constructed formal concept lattice encodes multi-level semantic relationships, allowing us to extract supervision signals at varying levels of generality. We begin by organizing attributes into sets on the basis of their level in the lattice. Specifically, for each $\mathcal{L}_i$, we identify the set of formal concepts residing at that level. To represent attributes corresponding to a certain semantic level of generality, we compute the union of *intents* (i.e., sets of attributes) associated with formal concepts in $\mathcal{L}_i$:

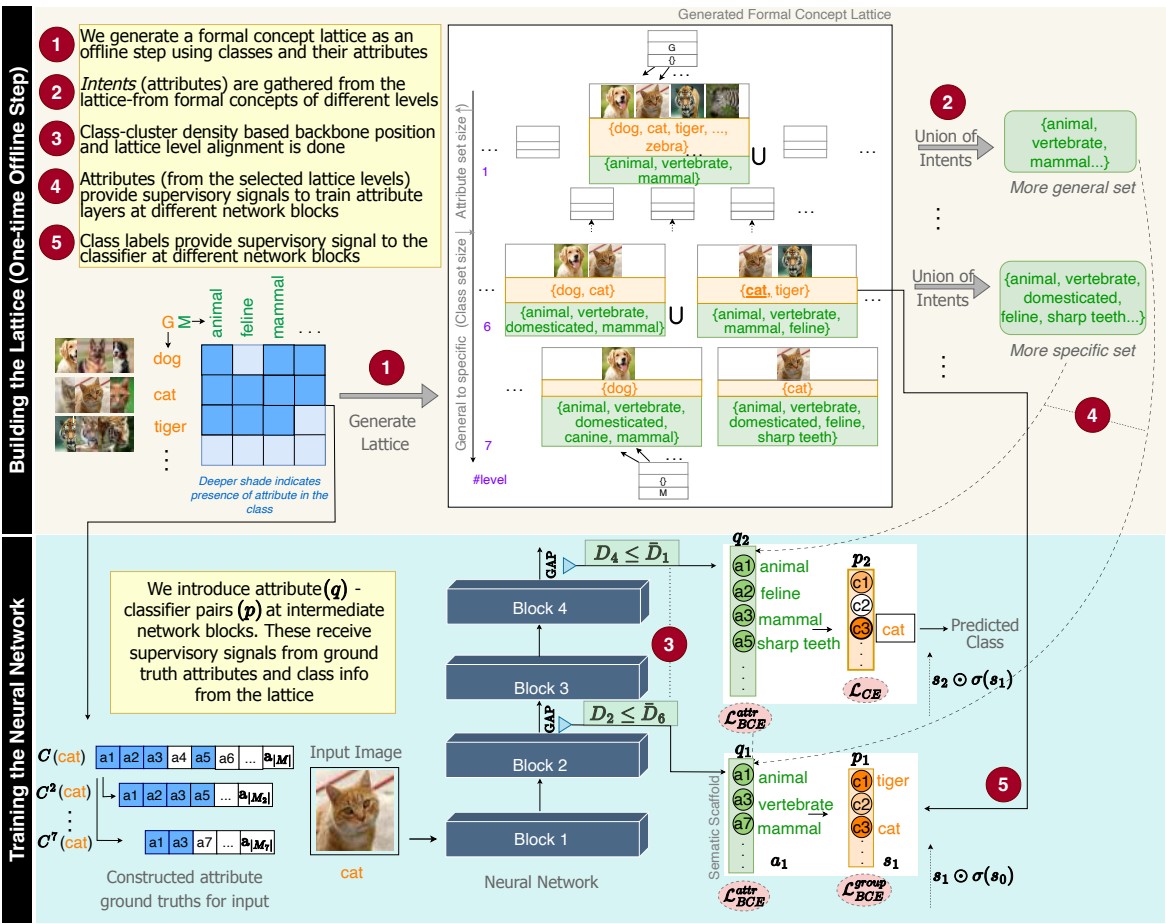

**Figure 2. Illustration of our overall approach.** A formal concept lattice is constructed for a concept-based setting using class and attribute sets and the binary relation between them *(top)*. A neural network is supervised at intermediate blocks using the information extracted from specific levels in the lattice *(bottom)*.

$$M_i = \bigcup_{\texttt{fc} \in \mathcal{L}_i} \texttt{fc.intent} \qquad (2)$$

where `fc.intent` denotes the *intent* of the formal concept. By repeating this process across different levels of the lattice, we obtain a hierarchy of attribute sets $\{M_1, M_2, \ldots, M_l\}$, each progressively more specific than the previous.

The concept lattice not only provides hierarchical attributes but also class groupings (*extents*) within each formal concept. We leverage this structure as a form of *supervision via iterative refinement*: we use loss functions at different layers such that they progressively narrow down the set of plausible classes (details in Sec 3.3). At early layers, general attribute sets may not identify individual classes but can often disambiguate groups of classes. For example, the presence of attributes such as {*whiskers, fur*} can rule out classes like *tortoise* or *whale*, narrowing the prediction space from all classes to, say, {*cat, dog, lion, tiger*}. This class group is then *cascaded* forward, where subsequent layers refine it using more specific attributes; the next layer may use *domesticated* to narrow this down to {*cat, dog*}.

Such a learning mechanism allows suppression of classes that get eliminated early, focusing the network's discriminative capacity on the remaining candidates. We theoretically show in Sec 4 that this iterative refinement mechanism preserves the semantic ordering imposed by the lattice structure, minimizing ordering violations during training (Sec 3.3).

**Class-Cluster Density.** Modern neural network architectures are organized into progressive modular components, for example, residual blocks in ResNets, transformer encoder blocks in ViTs, etc. We refer to these generically as 'blocks' and treat them as candidates for semantic supervision. We formalize the notion of *class-cluster density* to quantify the semantic granularity of intermediate network representations and to align lattice levels with network depth. Let $f_j(x) \in \mathbb{R}^{d_j}$ denote the feature embedding of input $x$ at network block $j$. Given a dataset with $n$ classes, we apply $k$-means clustering with $k = n$ to the set $\{f_j(x)\}$ over the training samples. For a cluster $K$, let $\mathcal{Y}(K)$ denote the set of ground-truth class labels of samples assigned to $K$. We define the class-cluster density at block $j$ as:

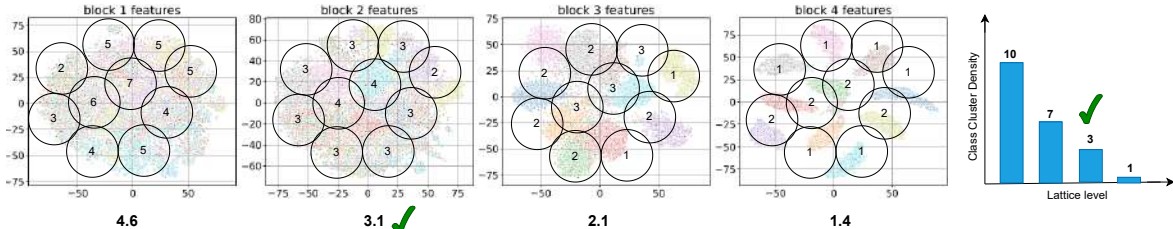

*Figure 3.* An illustrative example on 10 classes of how lattice levels are selected to supervise layers (blocks in our implementation) in the network. We perform clustering with $n$ (here 10) centers at each layer and obtain the average number of unique classes present per cluster, compute the same over the formal concept extents at each lattice level, and then choose closest alignment.

$$D_j = \frac{1}{n} \sum_{k=1}^{n} |\{\mathcal{Y}(K_k)\}| \qquad (3)$$

where $\{K_k\}_{k=1}^{n}$ are the resulting clusters. For example, if $K = \{(x1, y1), (x2, y2), (x3, y2)\}$, then $\mathcal{Y}(K) = \{y_1, y_2\}$ and $|\mathcal{Y}(K)| = 2$. The above equation averages this count across all $n$ clusters. Intuitively, $D_j$ measures the average number of distinct classes grouped together by the representation at depth $j$; higher values indicate more class-agnostic (general) features, while lower values reflect increased class-specific separation.

An analogous quantity can be defined for the formal concept lattice. For each lattice level $i$, we define this quantity as $\bar{D}_i = \frac{1}{|\mathcal{L}_i|} \sum_{fc \in \mathcal{L}_i} |\{\texttt{fc.extent}\}|$. By construction, higher lattice levels correspond to more general concepts and thus larger average extents, while lower levels correspond to finer-grained concepts with smaller extents. This establishes a natural alignment principle: early network blocks with high $D_j$ should be supervised by higher lattice levels with large $\bar{D}_i$, while deeper blocks with lower $D_j$ should align with lower lattice levels. In practice, we assign each lattice level $i$ to the earliest network block $j$ such that $D_j \leq \bar{D}_i$ (see Fig 2 bottom middle), ensuring that semantic supervision is applied at a representational depth whose granularity matches that of the corresponding lattice level. This strategy is illustrated in Figure 3 and an algorithm is included in Appendix A7.

### 3.3. FoCA CBM Training

We refer to our models as FoCA CBMs (Formal Concept Analysis CBMs). Once appropriate depths in the backbone have been determined, at each selected position $j$ in the network, we introduce a concept encoder $q_j$ to project the intermediate feature representation into a concept space defined by the attribute set $M_i$ from lattice level $i$ ($i$ denotes an index in the lattice, $j$ denotes an index in the network). Formally, each encoder learns a mapping $q_j : \mathbb{R}^{d_j} \rightarrow \mathbb{R}^{|M_i|}$ where $d_j$ is the dimensionality of the downsampled feature at layer $\hat{j}$, obtained via global average pooling, and $|M_i|$ is the size of the attribute set $M_i$ (see Fig 2, bottom). We also introduce $l$ classifiers $p_j : \mathbb{R}^{|M_i|} \rightarrow \mathbb{R}^n$, each operating on the output of the corresponding concept encoder $q_j$, where $n$ is the total number of classes.

**Obtaining Level-Wise Attribute and Class Ground Truths.** Given a sample $x$ with ground truth class label $y$, let $\mathbf{C}(x) \in \{0,1\}^{|M|}$ denote its complete ground truth attribute vector, where $|M|$ indicates all the attributes from the lattice. For lattice level $i$, let $\mathcal{M}_i \subseteq \{1, \ldots, |M|\}$ denote the set of attribute indices corresponding to attributes at that level. The level-wise ground truth attribute vector for sample $x$ at level $i$ is then obtained as $\mathbf{C}^i(x) = \mathbf{C}(x)[\mathcal{M}_i]$, i.e., by selecting the entries in $\mathbf{C}(x)$ indexed by attributes belonging to $\mathcal{M}_i$. Similarly, we define the *class group* of $y$ at level $i$ as the union of all class sets (extents) of formal concepts at $\mathcal{L}_i$ whose extents contain $y$. Formally, if $\mathcal{L}_i$ denotes the set of formal concepts at level $i$, then the class group is given by:

$$G_i(y) = \bigcup_{\substack{fc \in \mathcal{L}_i \\ y \in \texttt{fc.extent}}} \texttt{fc.extent} \qquad (4)$$

As an example, let level $l_i$ contains the formal concepts $\langle\{dog, cat\}, \{animal, vertebrate, ...\}\rangle$, $\langle\{cat, tiger\}, \{animal, vertebrate, ...\}\rangle$ and $\langle\{tiger, zebra\}, \{animal, vertebrate, ...\}\rangle$. If sample $x$ is a *cat*, since *cat* appears in two extents, the resulting group is $G_i(y) = \{dog, cat, tiger\}$. $\bar{G}_y^i = \mathbf{1}_{c \in G_y^i}$ represents the corresponding binary vector. This is Step 4 in Fig 2. As we proceed to deeper lattice levels, these groups become progressively smaller, ultimately converging to singleton sets representing individual classes.

**Overall Training Process.** The network is jointly trained for both attribute prediction and classification, using the aforementioned iterative refinement strategy. For each attribute encoder output, we apply a binary cross-entropy loss $\ell_{\text{BCE}}$ against the corresponding attribute set. For all intermediate classifiers, we supervise using group-level labels derived from the lattice, again using $\ell_{\text{BCE}}$. The final classifier is trained with standard cross-entropy loss $\ell_{\text{CE}}$ using the ground truth class label. Denoting $s_j(x) = p_j(q_j(f_j(x)))$ as the output of the classifier at layer $j$, we have $\hat{s}_j(x) = s_j(x) \cdot \sigma(s_{j-1}(x))$ as the post-iterative refinement output for classes at layer $j$, and $a_j(x) = \sigma(q_j(f_j(x)))$ as the sigmoid output for attributes at layer $j$. The overall loss $\ell_{\text{total}}$ is hence:

$$\alpha \sum_{j=1}^{l} (\underbrace{\mathbf{C}^i(x) \cdot \log(a_j(x)) + (1 - \mathbf{C}^i(x) \cdot (1 - \log(a_j(x))))}_{\ell_{\mathrm{BCE}_j}^{\mathrm{attr}}})$$

$$+ \beta \sum_{j=1}^{l} (\underbrace{\bar{G}_y^i \cdot \log(\hat{s}_j(x)) + (1 - \bar{G}_y^i) \cdot (1 - \log(\hat{s}_j(x)))}_{\ell_{\mathrm{BCE}_j}^{\mathrm{group}}})$$

$$+ \ell_{\mathrm{CE}_l} \quad (5)$$

where $\alpha$ and $\beta$ are weighting hyperparameters that balance the contribution of attribute and group-level supervision.

## 4. Some Theoretical Implications

We now present a theoretical analysis highlighting advantages of FoCA CBMs: (a) over non-hierarchical CBMs, and (b) more specifically, over other hierarchical CBM variants. Focusing first on (a), we study how imposing a lattice-ordered structure on classes and attributes provides a semantic scaffold for the network, in contrast to unordered alternatives, and show that FoCA CBMs preserve semantic concept ordering across layers. Let $f^{\mathrm{foca}}$ denote a FoCA CBM trained with iterative refinement , and let $f^{\mathrm{rnd}}$ denote a network trained with random class groupings $\{G_{\mathrm{rnd}}^i(y)\}$ that do not satisfy the subset constraint $G_{\mathrm{rnd}}^{i+1}(y) \subseteq G_{\mathrm{rnd}}^i(y)$. At zero training loss, preservation of ordering is trivial: a FoCA CBM respects lattice structure by construction, while random groupings may not. In general, multiple parameter configurations can attain the same empirical risk (Zhang et al., 2021). FoCA CBMs, however, introduce an inductive bias toward order-consistent solutions, ensuring that any formal concept activated at a given layer is also activated in all preceding layers, a guarantee absent under random group-based supervision. The substantive question lies in the realistic regime where the empirical risk $\hat{\ell}$ differs from the optimal risk $\ell^*$ by some margin $\epsilon > 0$. Crucially, not all prediction errors constitute ordering violations. For example, for a given input, if two consecutive layers incorrectly predict a true class as absent, the semantic ordering is preserved despite the misclassification. We therefore isolate and bound the probability of ordering-specific errors (configurations where $c \in \hat{G}_{i+1} \setminus \hat{G}_i$) as a function of the generalization gap $\epsilon = |\hat{\ell} - \ell^*|$. The following theorem establishes that, under bounded generalization error, a FoCA CBM maintains hierarchical consistency with high probability (over the training set $X$), while random supervision fails this guarantee even at zero training loss.

**Theorem 4.1** (Inductive Bias towards Order Consistency).
*For an input $(x, y) \sim X \times Y$, define:*

$$\hat{G}_i(x) = \{g \in G_i(y) \mid \hat{s}_j(x)[g] \geq \tau_c\},$$

*where $\tau_c \in [0, 1]$ is a threshold, typically chosen as $0.5$. Also, let $E_i^c$ denote the event $c \in \hat{G}_{i+1}(x) \setminus \hat{G}_i(x)$. Then, given $|\widehat{\ell_{total}} - \ell_{total}^*| \leq \epsilon$, assuming $\max_{i,c} P(E_{i-1}^c) + P(E_i^c) \leq \frac{1}{e}$,*

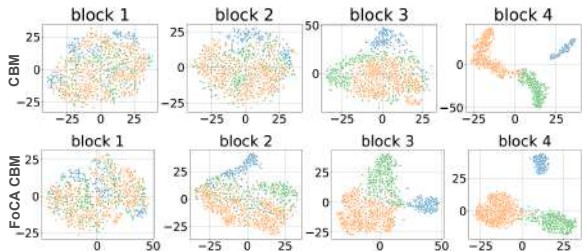

*Figure 4.* t-SNE plots of sample embeddings from AwA2 obtained from trained ResNet-18 backbones of Vanilla CBM and FoCA CBM. On a CBM, the clusters separate at the final block; in FoCA CBM, the separation happens gradually over blocks, showing graded semantic learning.

$$\Pr_{f^{\mathrm{foca}}} \left( \hat{G}_{i+1}(x) \not\subseteq \hat{G}_i(x) \right) \ll \Pr_{f^{\mathrm{rnd}}} \left( \hat{G}_{i+1}(x) \not\subseteq \hat{G}_i(x) \right).$$

We provide a brief proof sketch herein, while deferring the detailed proof to Appendix A3.

*Proof Sketch.* The proof proceeds in three steps. First, we establish that ground truth class groups respect the superset ordering: $G^{i+1}(y) \subseteq G^i(y)$ for all $i$ and $y$. Second, we quantify the loss penalty for ordering violations. Each violation at transition $i \to i+1$ incurs excess loss of at least $\gamma_{\min} > 0$ compared to correct predictions. Third, we apply the asymmetric Lovász Local Lemma (Erdős & Lovász, 1975) to the collection of violation events $\{E_i\}_{i=1}^{L-1}$. The dependency structure where each event $E_i$ depends only on adjacent events $E_{i-1}$ and $E_{i+1}$, enables us to lower bound the probability of maintaining ordering across all layers as $\Pr(\bigcap_i \overline{E}_i)$. In stark contrast, random groupings yield high violation probabilities, via a simple combinatorial argument, even at zero training loss. □

Notably this affinity for order consistency holds vacuously for vanilla CBMs where mappings are restricted to only the most fine-grained bottom layer of a formal concept lattice. FoCA CBMs, on the other hand, generalize this approach by allowing mappings to any layer of the formal concept lattice. We next analyze the advantages of using a formal concept lattice as opposed to an arbitrary hierarchy. Drawing on insights from the information bottleneck theory (Tishby & Zaslavsky, 2015), we posit that the increase in mutual information between successive layers and the output $I(f_j(X); Y) - I(f_{j-1}(X); Y)$ is upper-bounded by a constant $\Delta$, that depends on structural details such as the architecture and optimizer. In general, for DNNs, information gain between successive layers may be method-dependent and not guaranteed, as qualitatively illustrated by the t-SNE visualizations in the top row of Fig 4. In contrast, supervision of intermediate layers using a formal lattice induces a guaranteed lower bound on the information gain between any two consecutively supervised layers in FoCA-CBMs, a result formally established below in Theorem 4.2.

**Theorem 4.2** (Information-Theoretic Benefit of FCA Supervision). *Consider a FoCA-CBM trained with formal concept lattice $\mathcal{L}$ constructed from $\langle G, M, I \rangle$. Let network layer $j$ be*

*Table 1.* Results on Classification Test Accuracy, Cluster Impurity (CI) and Cluster Compactness (DBI) on ImageNet100, AwA2, CIFAR100 datasets averaged over 3 seeds. Best in bold, second best underlined. FoCA CBM-N is a Naive variant of our method where the attribute sets obtained from the lattice are all stacked after the backbone followed by a standard classifier.

| | ImageNet100 | | | AwA2 | | | CIFAR100 | | |
|---|---|---|---|---|---|---|---|---|---|
| | Acc ↑ | CI ↓ | DBI ↓ | Acc ↑ | CI ↓ | DBI ↓ | Acc ↑ | CI ↓ | DBI ↓ |
| **Vanilla CBM** [ICML'20] | 88.27±0.490 | 0.662±0.005 | 2.197±0.023 | 90.36±0.210 | 0.628±0.004 | 2.137±0.011 | 76.63±0.690 | 0.712±0.006 | 2.238±0.024 |
| **MLPCBM** | 86.88±0.290 | 0.659±0.006 | 2.210±0.029 | 89.65±0.370 | 0.637±0.007 | 2.120±0.018 | 76.17±0.620 | 0.722±0.006 | 2.276±0.027 |
| **Posthoc CBM** [ICLR'23] | 67.25±0.700 | 0.820±0.000 | 2.530±0.000 | 81.00±0.340 | 0.773±0.000 | 2.674±0.000 | 52.00±0.005 | 0.851±0.000 | 2.779±0.000 |
| **LFCBM** [ICLR'23] | 86.32±0.240 | 0.676±0.000 | 2.448±0.000 | 83.03±0.020 | 0.699±0.000 | 2.803±0.000 | 65.13±0.120 | 0.907±0.000 | 2.519±0.000 |
| **CEM** [NeurIPS'22] | 86.51±0.400 | 0.678±0.004 | 2.178±0.026 | **93.11**±**0.170** | 0.646±0.003 | 2.278±0.009 | 77.32±0.570 | 0.811±0.003 | 2.497±0.019 |
| **LaBo** [CVPR'23] | 74.48±0.004 | 0.676±0.000 | 2.448±0.000 | 91.53±0.010 | 0.699±0.000 | 2.803±0.000 | 65.23±0.003 | 0.907±0.000 | 2.519±0.000 |
| **SCBM** [NeurIPS'24] | 85.97±0.150 | 0.662±0.003 | 2.049±0.024 | 90.40±0.260 | 0.615±0.002 | 2.096±0.014 | 79.13±1.100 | 0.705±0.004 | 2.114±0.018 |
| **ProbCBM** [ICML'23] | 85.75±0.820 | 0.693±0.003 | 2.269±0.027 | 89.80±0.020 | 0.661±0.002 | 2.373±0.007 | 78.86±0.100 | 0.732±0.005 | 2.401±0.012 |
| **CF-CBM** [NeurIPS'24] | 87.30±0.010 | 0.676±0.000 | 2.448±0.000 | 89.19±0.810 | 0.699±0.000 | 2.803±0.000 | 60.02±0.540 | 0.907±0.000 | 2.519±0.000 |
| **HybridCBM** [CVPR'25] | 79.51±0.370 | 0.676±0.000 | 2.448±0.000 | 92.25±0.07 | 0.699±0.000 | 2.803±0.000 | 59.52±0.090 | 0.907±0.000 | 2.519±0.000 |
| **FoCA CBM-N** [Ours] | 88.36±0.290 | 0.665±0.005 | 2.150±0.021 | 88.26±0.270 | 0.659±0.005 | 2.273±0.015 | **82.41**±**0.050** | 0.688±0.005 | 2.177±0.021 |
| **FoCA CBM** [Ours] | **91.88**±**0.350** | **0.573**±**0.005** | **1.862**±**0.027** | 92.13±0.280 | **0.571**±**0.004** | 2.057±0.010 | 79.47±0.200 | **0.622**±**0.004** | **1.855**±**0.020** |

supervised by lattice level $i$ via class groups $G^i$ and attribute sets $M_i$. Then, under bounded training error $|\hat{\ell} - \ell^*| \leq \epsilon$ with $N$ training samples, the $\epsilon$-calibrated information gain of the network for layer $j$ is:

$$I_{\mathcal{D}}(f_j(X); Y) - I_{\mathcal{D}}(f_{j-1}(X); Y) \geq \Delta_{\text{lattice}}^{(i)} - 2\Delta_{\text{align}}(\epsilon),$$

where $\Delta_{\text{align}}(\epsilon) = O\left(\sqrt{\epsilon \log |G|} + N^{-1/2}\right)$.

The formality property (Eqn 1) guarantees that each attribute set $M_i$ is both informationally complete and parsimonious for its associated class-group structure. In contrast, random concept selection (Thm 4.1) breaks this optimality, resulting in ordering violations and unwarranted information loss.

## 5. Experiments

**Datasets:** We study our approach on three widely used benchmark datasets in concept-based learning: ImageNet100 (Russakovsky et al., 2015), AwA2 (Xian et al., 2019) and CIFAR100 (Krizhevsky et al., 2009). AwA2 is an expert-annotated dataset with class-level attributes, while for ImageNet100 and CIFAR100, we acquire class-level attribute annotations as in (Oikarinen et al., 2023) using an LLM. We use these benchmark datasets as they capture conceptual diversity among them. More dataset details are in Appendix A5.

**Baselines:** We compare our approach with ten concept-based learning models: (1) Vanilla CBMs (Koh et al., 2020), (2) MLPCBMs (an extension of vanilla CBMs), (3) Posthoc CBMs (Yuksekgonul et al., 2023), (4) Label-free CBMs (Oikarinen et al., 2023), (5) Concept Embedding Models (Zarlenga et al., 2022), (6) Language in a Bottle (Yang et al., 2023), (7) Stochastic CBMs (Vandenhirtz et al., 2024a), (8) Probabilistic CBMs (Kim et al., 2023), (9) Coarse-to-Fine CBMs (Panousis et al., 2024), and (10) Hybrid CBMs (Liu et al., 2025). These models represent different flavors of concept-based approaches with strong performance. All models, including ours, are ResNet-based.

**Metrics:** In addition to classification accuracy, we evaluate the semantic quality of learned embeddings across network depth using clustering-based metrics. After training, we apply $k$-means clustering with $n$ centers (= number of classes) to the feature embeddings at the end of each backbone block. We evaluate clusters using (i) Cluster Impurity (CI), using the Gini index (Breiman et al., 1984), which captures class heterogeneity within clusters, and (ii) Cluster Compactness, using the Davies–Bouldin Index (DBI) (Davies & Bouldin, 1979), which quantifies intra-cluster compactness and inter-cluster separation. Lower values of CI and DBI indicate more semantically structured representations.

**Results:** Our results over all metrics (*Test Accuracy, CI, DBI*) are reported for all baselines and our method in Table 1. FoCA CBM-N (N=Naive) is a naive variant of our method where attribute sets from multiple lattice levels are added as consecutive linear layers after the backbone. This is then trained like a Vanilla CBM with multiple attribute layers, followed by a classifier (one can view this as akin to a hierarchical CBM, with attribute sets constructed using our FCA lattice). The *CI* and *DBI* metrics give us a score per block of each model; we report the mean across all blocks. We see that our models consistently learn more meaningful embeddings, outperforming all baselines across datasets in terms of *CI* and *DBI*. In terms of accuracy, our models are consistently competitive across datasets.

**Performance on ViT Backbones:** Existing CBM methods largely focus on CNN architectures. To expand on this, we also studied our approach on ViT backbones, aligning appropriate levels from our lattice to intermediate blocks of a ViT architecture. We then compared our method (FoCA ViT) with a ViT CBM (CBM trained with a ViT backbone) on the CIFAR100 dataset. FoCA ViT outperformed the ViT CBM on all our metrics: $86.65_{\pm 0.295}$ vs $84.49_{\pm 0.547}$ test accuracy, $0.755_{\pm 0.004}$ vs $0.774_{\pm 0.016}$ CI and $1.983_{\pm 0.021}$ vs $2.237_{\pm 0.006}$ DBI, all averaged over three seeds (more

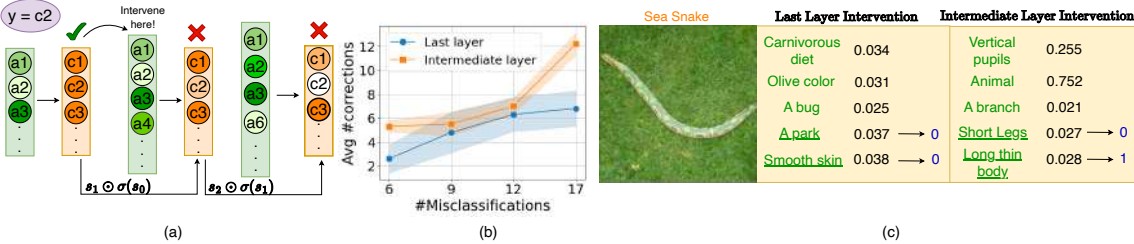

*Figure 5.* (a) Severity of misclassification informs which level of attributes to intervene on. Ground truth class for current input is $c2$, which is not in prediction set at layer 2; we hence intervene at 2nd layer; (b) Comparison of average number of corrections on interventions at last attribute layer (blue) and at appropriate intermediate layer on Imagenet100 (orange). Intervening at apt level improves performance; (c) Example attributes intervened on at last and intermediate layers; note that intermediate layers have more general attributes.

*Table 2.* Comparison with LLM (GPT4)-based hierarchy; both models show strong performance. However, turning off some attributes shows significant concept leakage in GPT4-based model, while FCA-based model shows strong concept compactness.

| Hierarchy | ImageNet100 | AwA2 | CIFAR100 |
|:---:|:---:|:---:|:---:|
| **FCA** | $91.88_{\pm0.35}$ $\downarrow$ $60.57_{\pm0.03}$ | $92.13_{\pm0.28}$ $\downarrow$ $91.51_{\pm0.29}$ | $79.47_{\pm0.20}$ $\downarrow$ $51.06_{\pm0.06}$ |
| **GPT4** | $91.47_{\pm0.41}$ $\downarrow$ $89.20_{\pm0.40}$ | $92.67_{\pm0.10}$ $\downarrow$ $92.30_{\pm0.39}$ | $80.35_{\pm0.07}$ $\downarrow$ $76.16_{\pm0.08}$ |

baselines in Appendix A4).

## 6. Analysis

**Considering Alternative Hierarchies:** Our framework is not restricted to FCA-derived hierarchies. Any hierarchical structure satisfying the subset-superset ordering property can be used to guide network learning. To demonstrate this generality, we propose an alternative LLM-based hierarchy construction method: An LLM (GPT4) is provided with the class-attribute incidence matrix and is prompted to generate attribute sets that satisfy the subset-superset relation, along with a group of classes associated with each class. This method produces a hierarchy with the required ordering properties and can be integrated into our framework. Table 2 (top value in each row) shows the competitive performance of the GPT4-based model, demonstrating our method's generalizability. That said, we observe that FCA-based hierarchies are superior in terms of *concept leakage*. To quantify this, we randomly *turn off* some attributes (1%) in both models and measure the resulting drop in test accuracy (standard test to study concept leakage in CBMs). We observe that the model based on GPT4 hierarchy shows minimal change, indicating significant concept leakage, while our FCA-based model shows strong concept compactness.

**Multi-Level Interventions:** A key advantage of CBMs is their support for test-time interventions that probe concept-to-class mappings. While standard approaches apply random interventions at the final concept layer, our framework enables *level-aware* interventions aligned with semantic granularity. Intuitively, coarse misclassifications

(e.g., *dog* vs. *elephant*) require intervention on general attributes, whereas fine-grained confusions (e.g., *dog* vs. *cat*) require more specific corrections. We quantify misclassification severity by identifying the deepest layer at which the ground-truth class remains in the predicted class group (Fig 5(a)). Due to iterative refinement, classes eliminated at a given level rarely reappear, making this layer a natural intervention point. We intervene at the corresponding attribute layer by randomly modifying $k$ attributes and propagating the updated activations forward, and compare this to interventions applied only at the final layer. As shown in Fig 5(b), level-aware interventions consistently outperform final-layer interventions across four random sets of misclassifications on ImageNet100 (averaged over 10 trials of 20 interventions per sample). Fig 5(c) illustrates that earlier layers predominantly involve general attributes, whereas final-layer interventions mix general and fine-grained concepts.

Additionally, we evaluate our models intervention effectiveness with respect to random oracle interventions over the whole test set and find that

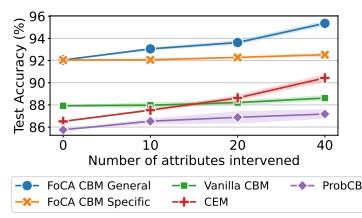

our general-layer interventions yield the highest accuracy gains per oracle intervention, as is shown in the oracle interventions vs accuracy gains curve to the right. Together, these results show that semantic scaffolding enables effective interventions at multiple abstraction levels.

**Scaling and Practicality:** Formal concept lattices are constructed as a one-time offline preprocessing step, taking approximately 4.5 seconds for ImageNet100, 37 seconds for AwA2, and 2 seconds for CIFAR100.

*Table 3.* Computational cost (in Giga Floating Point Operations) of different models on all three datasets.

| Model | ImageNet100 | AwA2 | CIFAR100 |
|:---:|:---:|:---:|:---:|
| **CBM** | 8.21G | 3.64G | 8.21G |
| **CEM** | 8.259G | 3.64G | 8.26G |
| **SCBM** | 17.44G | 7.28G | 17.44G |
| **FoCA CBM** | 8.21G | 3.64G | 8.21G |

The worst-case complexity of lattice construction is

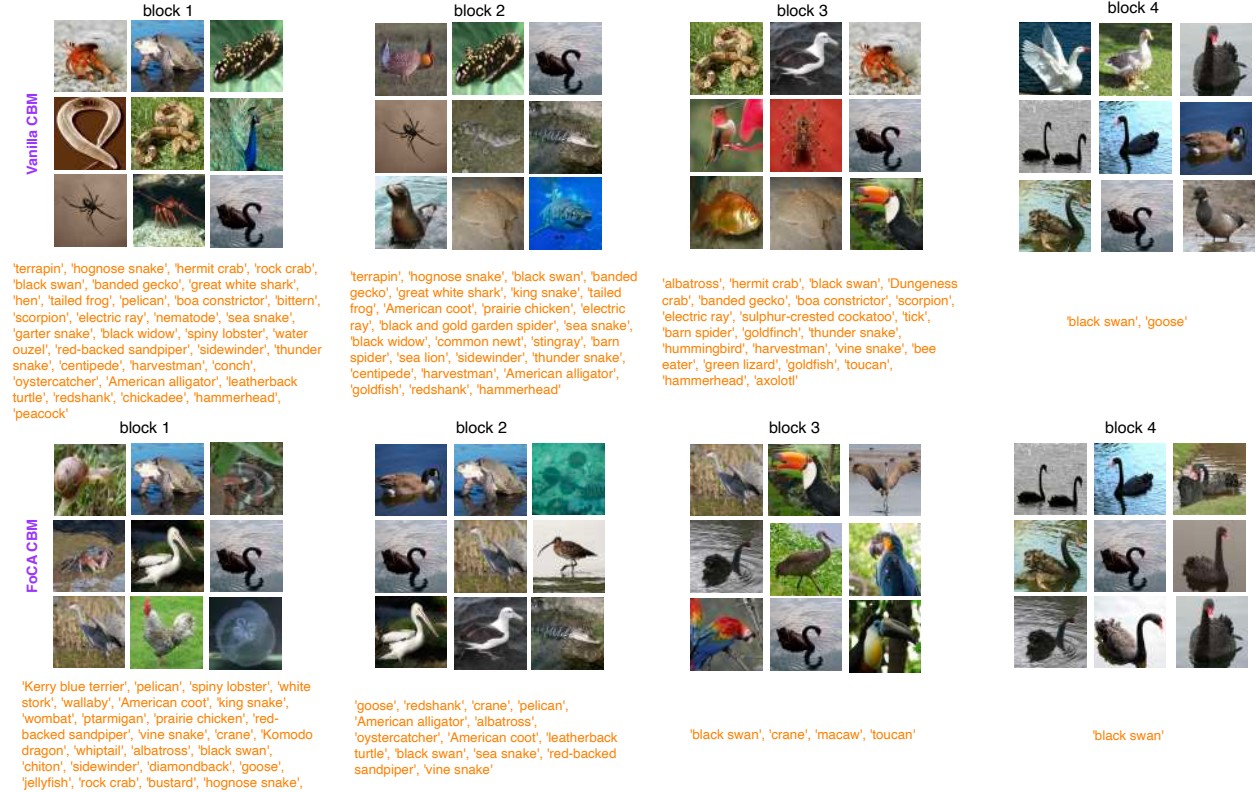

Figure 6. **Visualizing some class clusters at different blocks**. Classes from the clusters of a Vanilla CBM are mixed up across blocks, whereas a FoCA CBM gradually telescopes relevant concepts.

$O(|\mathcal{L}| \cdot |G|^2 \cdot |M|)$, where $|\mathcal{L}|$ is the number of formal concepts; however, real-world class–attribute relations are typically sparse (fill ratio $< 0.1$), resulting in near-quadratic growth in practice (Table A12). Table 3 reports the computational cost (in FLOPs) of FoCA CBMs and representative baselines. Despite introducing intermediate supervision, FoCA CBMs incur computational costs comparable to standard CBMs. Prior work has also shown that formal concepts remain stable under incremental updates (Kuznetsov, 2007), supporting the applicability of FCA-based hierarchies in dynamic settings.

**Looking into the Clusters:** We examine the clusters from the blocks of a FoCA CBM and Vanilla CBM. As shown in Fig 6, the classes in a Vanilla CBM's clusters only separate out into the group {*black swan, goose*} at the last block, being quite noisy throughout. However in a FoCA CBM, we observe gradual semantic refinement. The first block captures a mixture of classes, the second block refines it to classes that are associated with water, the third block narrows this down to a set of birds and finally the last block has a separate cluster for {*black swan*}.

Further analysis, ablations, and more results including interventions, impact of iterative refinement, lattice and position choices are provided in Appendix A2, A4.

## 7. Conclusion

In this work, we examine how concept-based learning models can respect hierarchical structure across a network's depth. Rather than learning concepts as a flat set at a single stage, we propose guiding representation learning using a structured semantic hierarchy derived from Formal Concept Analysis. The resulting formal concept lattice provides principled supervision points that align semantic granularity with network depth, enabling representations to evolve naturally from general to specific. Our approach improves interpretability by exposing meaningful intermediate concepts and enhances learning through gradual semantic refinement, as reflected in improved clustering quality. We further show that multi-level interventions are both feasible and more effective than standard single-layer alternatives through our approach. We believe this framework brings concept-based models closer to human semantic reasoning, and opens avenues for weakly supervised hierarchies and extensions to other modalities.

**Acknowledgments:** Deepika SN Vemuri and Krishn Vishwas Kher would like to thank the MoE Prime Minister's Research Fellowship (PMRF) for the fellowship support. We thank the anonymous reviewers for their helpful feedback in improving the presentation of the paper.

## Impact Statement

Our work advances the fields of Concept-Based Learning and Interpretability by establishing a principled method for organizing semantic concepts in neural networks. The hierarchical structure we introduce enables multi-level interpretability, where users can interact with models at varying levels of abstraction. We believe that this has significant implications for high-stakes domains like healthcare and medical diagnostics where the operators can verify that a model's reasoning respects the appropriate diagnostic hierarchies (e.g. identifying organs before specific pathologies). The multi-level intervention capability enables more nuanced interactions with these models. Users can now target corrections at the appropriate level of semantic granularity - a broad category mistake or finer-grained mistake. By formalizing the connection between Formal Concept Analysis and Deep Learning, our work bridges communities that have traditionally worked in isolation. We believe that this cross-pollination could inspire new research directions leading to models that better align with human cognitive structures.

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

# Appendix

In this appendix, we provide additional details of our work, including the following information.

## Table of Contents

## A1. Notation

| Symbol | Description |
|---|---|
| $\langle G, M, I \rangle$ | Formal context of the set of objects $G$, the set of attributes $M$, with the incidence relation $I$ |
| $\langle A, B \rangle$ | Formal concept with the extent $A \subseteq G$ and the intent $B \subseteq M$ |
| $\uparrow, \downarrow$ | Concept-forming operators |
| $\preceq$ | Subset-superset ordering |
| $\mathcal{B}(G, M, I)$ | Set of all formal concepts corresponding to the formal context $\langle G, M, I \rangle$ |
| $\mathcal{L}$ | Formal concept lattice, $\mathcal{L} = \langle \mathcal{B}(G, M, I), \preceq \rangle$ |
| $L$ | Number of levels in the lattice |
| $\mathcal{L}_1, \dots, \mathcal{L}_L$ | Set of formal concepts at each level of the lattice |
| $M_1, \dots, M_L$ | Attribute set at each level of the lattice |
| $n$ | Total number of classes |
| $\sigma$ | Sigmoid |
| $d_j$ | Dimensionality of the downsampled feature space at selected position $j$ in the backbone network |
| $f_j$ | Global average pooling output at semantic layer $j$ for input $x$, $f_j : \mathcal{X} \rightarrow \mathbb{R}^{d_j}$ |
| $q_j$ | Concept encoder for semantic layer $j$ corresponding to lattice layer $i$, $q_j : \mathbb{R}^{d_j} \rightarrow \mathbb{R}^{|M_i|}$ |
| $p_j$ | Classifier for semantic layer $j$ corresponding to lattice layer $i$, $p_j : \mathbb{R}^{|M_i|} \rightarrow \mathbb{R}^n$ |
| $s_j$ | Classifier output at semantic layer $j$ for input $x$, $s_j : \mathcal{X} \rightarrow \mathbb{R}^n = p_j \circ q_j \circ f_j$ |
| $\hat{s}_j$ | Post-iterative refinement output at semantic layer $j$ for input $x$, $\hat{s}_j : \mathcal{X} \rightarrow \mathbb{R}^n = s_j \cdot (\sigma \circ s_{j-1})$ |
| $a_j$ | Attribute sigmoid output at semantic layer $j$ for input $x$, $a_j : \mathcal{X} \rightarrow \mathbb{R}^n = \sigma \circ q_j \circ f_j$ |
| $\mathbf{C}$ | Complete ground truth attribute vector for sample $x$ |
| $\mathbf{C}^i$ | Ground truth attribute vector for sample $x$ at lattice level $i$ |
| $G_y^i$ | Class group of sample $x$ with class label $y$ at lattice level $i$ |
| $H(\cdot)$ | Entropy |
| $I_{\mathcal{D}}(\cdot; \cdot)$ | Mutual information for dataset $\mathcal{D}$ |

## A2. Ablations

**Ablation on Iterative Refinement:** We study the impact of *iterative refinement* and perform an ablation with and without it on our method. We see a consistent drop in accuracy across datasets without iterative refinement, with consistent increases in *CI* and *DBI* as well, on almost all datasets. We report these results in Tables A4, A5 and A6. These results highlight the usefulness of our class group-based refinement strategy.

*Table A4.* Our group-based iterative refinement strategy improves on all metrics on CIFAR100.

| | CIFAR100 | | |
|---|---|---|---|
| | Acc $\uparrow$ | CI $\downarrow$ | DBI $\downarrow$ |
| With | 79.7 | 0.6221 | 1.8558 |
| Without | 77.19 | 0.6612 | 1.8762 |

*Table A5.* Our group-based iterative refinement strategy improves on all metrics on ImageNet100.

| | ImageNet100 | | |
|---|---|---|---|
| | Acc $\uparrow$ | CI $\downarrow$ | DBI $\downarrow$ |
| With | 91.92 | 0.5737 | 1.8623 |
| Without | 88.19 | 0.6162 | 2.0639 |

**Ablation on Class-Cluster Density:** We study the impact of class-cluster-density-based backbone position selection and lattice level selection. In order to isolate the impacts, we try both (1) fixing the backbone position and using non class-cluster-density-based lattice level selection, and (2) fixing the lattice level and using non class-cluster-density-based

*Table A6.* Our group-based iterative refinement strategy improves on almost all metrics on AwA2, with the CI scores being in the same range.

| | AwA2 | | |
| --- | --- | --- | --- |
| | Acc ↑ | CI ↓ | DBI ↓ |
| With | 92.20 | 0.5288 | 1.932 |
| Without | 91.85 | 0.5231 | 2.017 |

backbone position selection. We use two semantic layers for these experiments and provide the results in Tables A7 and A8. The results indicate how different levels carry different amounts of information.

*Table A7.* Accuracy on varying the semantic layers position in FoCA CBM models on all datasets. Models have the following naming scheme: `FoCA_<levels>_<positions>`. For example, FoCA_(2, 1)_(3, 4) indicates that the attributes are extracted from the 2nd and 1st levels in the lattice, and are used to supervise the 3rd and 4th blocks in the ResNet. clf0 and clf1 correspond to the two classifiers from the two semantic layers.

| Model | AwA2 | | Model | CIFAR100 | | Model | ImageNet100 | |
| --- | --- | --- | --- | --- | --- | --- | --- | --- |
| | clf0 | clf1 | | clf0 | clf1 | | clf0 | clf1 |
| FoCA_(2, 1)_(2, 4) | 85.61 | 89.88 | FoCA_(2, 1)_(2, 4) | 92.41 | 68.89 | FoCA_(5,1)_(2, 4) | 79.56 | 90.69 |
| FoCA_(2, 1)_(3, 4) | 87.90 | 92.20 | FoCA_(2, 1)_(3, 4) | 94.30 | 79.70 | FoCA_(5,1)_(3, 4) | 96.26 | 91.92 |

*Table A8.* Accuracy on varying the levels chosen from the lattice in FoCA CBM models on all datasets. Models have the following naming scheme: `FoCA_<levels>_<positions>`. For example, FoCA_(3, 1)_(3, 4) indicates that the attributes are extracted from the 3rd and 1st levels in the lattice, and are used to supervise the 3rd and 4th blocks in the ResNet. clf0 and clf1 correspond to the two classifiers from the two semantic layers.

| Model | AwA2 | | Model | CIFAR100 | | Model | ImageNet100 | |
| --- | --- | --- | --- | --- | --- | --- | --- | --- |
| | clf0 | clf1 | | clf0 | clf1 | | clf0 | clf1 |
| FoCA_(3, 1)_(3, 4) | 87.90 | 92.20 | FoCA_(2, 1)_(3, 4) | 94.30 | 79.70 | FoCA_(3,1)_(3, 4) | 91.80 | 88.30 |
| FoCA_(4, 1)_(3, 4) | 95.40 | 91.61 | FoCA_(3, 1)_(3, 4) | 92.16 | 78.44 | FoCA_(5,1)_(3, 4) | 96.26 | 91.92 |
| FoCA_(6, 1)_(3, 4) | 98.27 | 91.77 | FoCA_(5, 1)_(3, 4) | 89.26 | 73.91 | FoCA_(7,1)_(3, 4) | 93.98 | 91.00 |

## A3. Proofs and Further Theoretical Analysis

In this section, we provide the proof of Theorem 1, along with a similar theorem for attributes.

**Theorem 4.1** (Inductive Bias towards Order Consistency). *For an input $(x, y) \sim X \times Y$, define:*

$$\hat{G}_i(x) = \{g \in G_i(y) \mid \hat{s}_j(x)[g] \geq \tau_c\},$$

*where $\tau_c \in [0, 1]$ is a threshold, typically chosen as $0.5$. Also, let $E_i^c$ denote the event $c \in \hat{G}_{i+1}(x) \setminus \hat{G}_i(x)$. Then, given $|\widehat{\ell_{total}} - \ell_{total}^*| \leq \epsilon$, assuming $\max_{i,c} P(E_{i-1}^c) + P(E_i^c) \leq \frac{1}{e}$,*

$$\Pr_{f^{foca}} \left( \hat{G}_{i+1}(x) \not\subseteq \hat{G}_i(x) \right) \ll \Pr_{f^{rnd}} \left( \hat{G}_{i+1}(x) \not\subseteq \hat{G}_i(x) \right).$$

*Proof.* We provide mathematical details to the proof sketch described in Section 4 of the main paper. We begin by proving the following simple lemma.

**Step 1: Ground Truth Ordering.**

**Lemma A3.1.** $G^{i+1}(y) \subseteq G^i(y)$ *for all $i \in [L-1]$, $y \in G$.*

*Proof.* For sake of contradiction, suppose $\exists c \in G^{i+1}(y) \setminus G^i(y)$. Then $\exists \langle A, B \rangle \in \mathcal{L}_{i+1}$ with $c, y \in A$. By lattice structure, $\exists \langle C, D \rangle \in \mathcal{L}_i$ with $A \subseteq C$ (subconcept property). Thus $c \in C$ and $y \in C$, implying $C \subseteq G^i(y)$ by definition, giving $c \in G^i(y)$, which is absurd. Hence, the assertion in the lemma holds. $\square$

---

**Algorithm 1** Training Lattice-Guided Concept-Based Models

---

**Require:** Dataset $\mathcal{D} = \{(x, c, y)\}$, class-attribute annotations $(G, M, I)$, base network $f$, number of supervision points $l$, loss weights $\alpha, \beta$

1: $\mathcal{L} \leftarrow \text{CONSTRUCTLATTICE}(G, M, I)$           ▷ Build concept lattice
2: **for** $i = 1$ to $l$ **do**
3:    $\mathcal{M}_i \leftarrow \bigcup_{\langle A,B \rangle \in \mathcal{L}_i} B$       ▷ Extract attribute set from level $i$ of the lattice $\mathcal{L}$
4: **end for**
5: $\{j_1, \ldots, j_l\}, \{\mathcal{L}_1, \ldots, \mathcal{L}_l\} \leftarrow \text{SELECTLAYERSANDLEVELS}(f, \mathcal{D}, \mathcal{L}, l)$   ▷ Select supervision points in network and lattice levels using class cluster density
6: Initialize concept encoders $\{q_1, \ldots, q_l\}$ and classifiers $\{p_1, \ldots, p_l\}$
7: **for** epoch $= 1$ to $N$ **do**
8:    **for** each $(x, c, y) \in \mathcal{D}$ **do**
9:      $\hat{y}_0 \leftarrow \infty$             ▷ Initialize with all classes possible
10:      $\ell_{\text{attr}} \leftarrow 0, \ell_{\text{group}} \leftarrow 0, \ell_{\text{class}} \leftarrow 0$
11:      **for** $i = 1$ to $l$ **do**
12:        $h_i \leftarrow f_{\leq j_i}(x)$         ▷ Forward through layer $j_i$
13:        $\hat{c}_i \leftarrow q_i(h_i)$         ▷ Predict concepts from $M_i$
14:        $\hat{y}_i \leftarrow p_i(\hat{c}_i)$          ▷ Predict classes
15:        $C_y^i \leftarrow [c[\mathcal{M}_i]]$ ▷ Ground truth concepts for level $i$ by selecting the concepts from $c$ that are present in $\mathcal{M}_i$
16:        $G_y^i \leftarrow \bigcup_{\langle A,B \rangle \in \mathcal{L}_i, y \in A} A$        ▷ Ground truth group
17:        $\ell_{\text{attr}} \leftarrow \ell_{\text{attr}} + \text{BCE}(\sigma(\hat{c}_i), C_y^i)$        ▷ Concept loss
18:        $\hat{y}_i \leftarrow \hat{y}_i \odot \sigma(\hat{y}_{i-1})$        ▷ Iterative refinement
19:        **if** $i < l$ **then**
20:          $\ell_{\text{group}} \leftarrow \ell_{\text{group}} + \text{BCE}(\sigma(\hat{y}_i), G_y^i)$        ▷ Group loss
21:        **else**
22:          $\ell_{\text{class}} \leftarrow \text{CE}(\text{SOFTMAX}(\hat{y}_i), y)$        ▷ Final class loss
23:        **end if**
24:      **end for**
25:      $\ell_{\text{total}} \leftarrow \alpha \cdot \ell_{\text{attr}} + \beta \cdot \ell_{\text{group}} + \ell_{\text{class}}$
26:      Update parameters via backpropagation on $\ell_{\text{total}}$
27:    **end for**
28: **end for**
29: **return** Trained model with hierarchical concept structure

---

**Step 2: Loss Penalty for Violations.**

Next, we examine how much extra loss is incurred by an ordering mismatch over a prediction that obeys the ground truth, to later help us derive a bound on the probability of ordering mismatches. For simplicity, we assume the $0 - 1$ loss, and we only consider the group loss for the rest of this proof. First, note that for a sample $(x, y)$ and class $c$, the per-layer loss is:

$$\ell_i^c = \begin{cases} -\log \hat{y}_i^c & \text{if } c \in G^i(y) \\ -\log(1 - \hat{y}_i^c) & \text{if } c \notin G^i(y) \end{cases} \tag{6}$$

**Lemma A3.2.** *If $c \in \hat{G}_{i+1}(x) \setminus \hat{G}_i(x)$ (ordering violation), then:*

$$\ell_i^c + \ell_{i+1}^c \geq \ell_i^{c,*} + \ell_{i+1}^{c,*} + \gamma_{\min} \tag{7}$$

*where $\ell_i^{c,*}$ is the optimal loss achievable respecting ordering, and $\gamma_{\min} = -2 \max(\log(\tau), \log(1 - \tau))$.*

*Proof.* Since neural networks are universal function approximators, we take $\ell_i^{c,*} = 0$. By Lemma A3.1, there are three exhaustive cases to consider, namely:

- *Case 1:* $c \in G^{i+1}(y) \subseteq G^i(y)$. Ground truth requires $\hat{y}_i^c, \hat{y}_{i+1}^c \geq \tau$.

- *Case 2:* $c \in G^i(y) \setminus G^{i+1}(y)$. Ground truth requires $\hat{y}_i^c \geq \tau$, $\hat{y}_{i+1}^c < \tau$.

- *Case 3:* $c \notin G^i(y) \cup G^{i+1}(y)$. Ground truth requires both predictions low.

Simple casework in each of these cases gives $\gamma_{\min} = -2 \max(\log(\tau), \log(1 - \tau))$.

$\square$

**Step 3: Probability Upper Bound For a Single Ordering Mismatch.**

By the definition of the group loss function and its contribution to the total loss (Objective 5), we know that the risk $\widehat{\ell_{\text{group}}} \leq \widehat{\ell_{\text{total}}}$. Furthermore, we assume that $\ell_{\text{total}}^* = \ell_{\text{group}}^* = 0$ owing to the universal function approximation capabilities of neural networks. Therefore, given that $|\widehat{\ell_{\text{total}}} - \ell_{\text{total}}| \leq \epsilon$, we can sum individual risks specific to layer indices and classes to write,

$$|\widehat{\ell_{\text{total}}} - \ell_{\text{total}}| \leq \epsilon \iff \sum_{i=1}^{L-1} \sum_{c \in G} \ell_i^c + \ell_{i+1}^c \leq \sum_{i=1}^{L-1} \sum_{c \in G} \ell_i^{c,*} + \ell_{i+1}^{c,*} + 2\epsilon.$$

Next, we write out the sum of losses at two consecutive layers for the same class as a total expectation over the probability of the event $E_i^c$

$$\mathbb{E}_{(x,y)\sim\mathcal{D}} \left[ \sum_{i=1}^{L-1} \sum_{c \in G} \ell_i^c + \ell_{i+1}^c - \ell_i^{c,*} - \ell_{i+1}^{c,*} \right] \leq 2\epsilon. \tag{8}$$

Switching sums and expectations, we get:

$$\mathbb{E}_{(x,y)\sim\mathcal{D}} \left[ \sum_{i=1}^{L-1} \sum_{c \in G} \ell_i^c + \ell_{i+1}^c - \ell_i^{c,*} - \ell_{i+1}^{c,*} \right] = \sum_{i=1}^{L-1} \sum_{c \in G} \mathbb{E}_{(x,y)\sim\mathcal{D}} \left[ \ell_i^c + \ell_{i+1}^c - \ell_i^{c,*} - \ell_{i+1}^{c,*} \right] \geq$$
$$\sum_{i=1}^{L-1} \sum_{c \in G} \mathbb{P}(E_i^c) \cdot \gamma_{\min} \implies \sum_{i=1}^{L-1} \sum_{c \in G} \mathbb{P}(E_i^c) \leq \frac{2\epsilon}{\gamma_{\min}}, \tag{9}$$

where the first inequality follows by expanding the inner expectation of the Bernoulli random variable $E_i^c$ and the fact from Lemma A3.2 that $\ell_i^c + \ell_{i+1}^c - \ell_i^{c,*} - \ell_{i+1}^{c,*} \geq \gamma_{\min}$, since $\ell_i^c + \ell_{i+1}^c$ is a non-negative cross entropy loss and $\ell_i^{c,*} + \ell_{i+1}^{c,*}$ is 0.

**Step 4: Applying Asymmetric Lovász Local Lemma.**

To provide a tighter lower bound on the probability that no ordering mismatch occurs under lighter assumptions, we turn to applying the (asymmetric version of the) Lovász Local Lemma (*LLL*) to the events $E_i^c$. We begin by stating the *LLL* as follows (Harvey & Vondrák, 2015; Erdos & Lovász, 1975):

**Lemma A3.3** (*Asymmetric Lovász Local Lemma*). *Let $\mathcal{E}_1, \mathcal{E}_2, \ldots, \mathcal{E}_n$ be events in a probability space with a dependency graph $\Gamma$ such that $\mathbb{P}(\mathcal{E}_i) \leq p_i \in [0, 1]$. If $\exists y_1, y_2, \ldots y_n > 0$ for which:*

$$p_i \leq \frac{y_i}{\sum_{S \subseteq \Gamma^+(i)} \Pi_{j \in S}(1 + y_j)}, \ \forall i \in [n], \tag{10}$$

*then $\mathbb{P}\left( \bigcap_{i=1}^n \overline{\mathcal{E}_i} \right) = \Pi_{i \in [n]} \frac{1}{(1+y_i)} > 0$.*

Here, $\Gamma^+(i)$ denotes the union of the neighbors of the event $\mathcal{E}_i$ (along with itself) in $\Gamma$, the dependency graph that exists over the events. In our setting, the only neighbour that a given event $\mathcal{E}_i^c$ directly depends on is $\mathcal{E}_{i-1}^c$ for $i > 0$. Therefore applying the *LLL* in our setting reduces to finding $y_1, y_2, \ldots y_{|G| \times L}$ such that,

$$\Pr(E_i^c) \leq \frac{y_{\{c,i\}}}{(1 + y_{\{c,i\}})(1 + y_{\{c,i-1\}})}, \ \forall i \in [n] \setminus \{0\}, c \in G. \tag{11}$$

To do so, we first note that since the events $E_i^c$ correspond to Bernoulli random variables, we can apply a tight inequality applicable for Bernoulli random variables known as the Kearns-Saul inequality (Kearns & Saul, 1998; Berend & Kontorovich, 2015), stated as follows:

**Lemma A3.4** (*Kearns-Saul Inequality*). *For all $p \in (0,1)$ and $t \in \mathbb{R}$,*

$$(1-p)e^{-tp} + pe^{t(1-p)} \leq e^{\left(\frac{1-2p}{4\log\left(\frac{1-p}{p}\right)}t^2\right)}. \tag{12}$$

The above inequality can be rewritten as,

$$p \leq \frac{e^{\left(\frac{1-2p}{4\log\left(\frac{1-p}{p}\right)}\right)t^2 + pt} - 1}{(e^t - 1)}, \ \forall p \in (0,1), t \in \mathbb{R}. \tag{13}$$

Notice that $f(t) = \left(\frac{1-2p}{4\log\left(\frac{1-p}{p}\right)}\right)t^2 + pt - t$ attains its minimum value at $t^*(p) = 2 \cdot \frac{1-p}{1-2p}\log(\frac{1-p}{p})$. Furthermore, $f(t^*(p)) \leq 0 \leq t^*(p)$, as can be verified by direct calculation. Therefore for a fixed $p \in (0,1)$ we can write,

$$p \leq \frac{e^{f(t^*(p))+t^*(p)} - 1}{(e^{t^*(p)} - 1)} = \frac{e^{f(t^*(p))+t^*(p)} - e^{f(t^*(p))} + e^{f(t^*(p))} - 1}{(e^{t^*(p)} - 1)} = e^{f(t^*(p))} + \frac{e^{f(t^*(p))} - 1}{(e^{t^*(p)} - 1)} \leq e^{f(t^*(p))}. \tag{14}$$

Now let $\kappa = e$. Choose $y_{\{c,i\}} = e^{\kappa P(E_i^c)} - 1$. Notice that,

$$\frac{y_{\{c,i\}}}{(1 + y_{\{c,i\}})(1 + y_{\{c,i-1\}})} \tag{15}$$

$$= e^{-\kappa(\mathbb{P}(E_{i-1}^c)+\mathbb{P}(E_i^c))} \cdot \left(e^{\kappa \mathbb{P}(E_i^c)} - 1\right) \tag{16}$$

$$\geq e^{-\kappa(\mathbb{P}(E_{i-1}^c)+\mathbb{P}(E_i^c))} \cdot \left(e \cdot \mathbb{P}(E_i^c)\right), \tag{17}$$

where the last inequality can be obtained by first noting that the functions $h(x) = e^{\kappa x} - 1$ and $g(x) = \kappa x$ are monotonic functions on $[0,1]$ with $h(0) = g(0) = 0$. Writing the Taylor series expansion of $h(x), g(x)$ in $[0,1]$ shows that proving $\frac{\partial h}{\partial x} \geq \frac{\partial g}{\partial x} \ \forall x \in [0,1] \implies h(x) \geq g(x) \ \forall x \in [0,1]$. Consequently, we note that $\frac{\partial h}{\partial x} = \kappa e^{\kappa x}$, and $\frac{\partial g}{\partial x} = \kappa$. Lastly, note that $\min_{[0,1]} \frac{\partial h}{\partial x} = \kappa$, which proves the required assertion.

By using our assumption that $\mathbb{P}(E_i^c) + \mathbb{P}(E_{i-1}^c) \leq \frac{1}{e}$, we have that,

$$1 - \kappa(\mathbb{P}(E_i^c) + \mathbb{P}(E_{i-1}^c)) \geq 0 \tag{18}$$

by direct substitution.

Therefore,

$$\mathbb{P}(E_i^c) \leq \frac{y_{\{c,i\}}}{(1 + y_{\{c,i\}})(1 + y_{\{c,i-1\}})}, \ \forall i \in [n] \setminus \{0\}, c \in G. \tag{19}$$

Therefore, $\mathbb{P}(\bigcap_{i=1}^{|G|\cdot L} \overline{\mathcal{E}_i}) = \Pi_{i\in[|G|\cdot L]}\frac{1}{(1+y_i)} \geq \Pi_{i\in[|G|\cdot L]}e^{-\kappa\mathbb{P}(E_i^c)} \geq e^{-\frac{2\epsilon\kappa}{\gamma_{\min}}}.$ $\qquad \square$

**Step 5: Random Baseline Analysis.**

**Lemma A3.5.** *For random groupings where $G_{rnd}^{i+1}(y) \not\subseteq G_{rnd}^i(y)$ in general, with group sizes $k_{i+1}, k_i$:*

$$\Pr(E_i^c) \geq \frac{k_{i+1}}{|G|}\left(1 - \frac{k_i}{|G|}\right) \tag{20}$$

*For hierarchies with $k_i = \Theta(|G|/L)$, this gives $\Pr(E_i^c) = \Omega(1/L)$.*

*Proof.* Under random independent assignment, a class $c$ satisfies:

$$\Pr(c \in G_{\text{rnd}}^{i+1}(y)) = \frac{k_{i+1}}{|G|} \tag{21}$$

$$\Pr(c \notin G_{\text{rnd}}^{i}(y)) = 1 - \frac{k_i}{|G|} \tag{22}$$

Therefore:

$$\Pr(c \in G_{\text{rnd}}^{i+1}(y) \setminus G_{\text{rnd}}^{i}(y)) = \frac{k_{i+1}}{|G|} \left(1 - \frac{k_i}{|G|}\right) \tag{23}$$

The probability of at least one violation among $|G|$ classes:

$$\Pr(E_i^c) = 1 - \left(1 - \frac{k_{i+1}}{|G|} \left(1 - \frac{k_i}{n}\right)\right)^{|G|} \tag{24}$$

In our experiments typical hierarchies are about $k_{i+1} = |G|/(2L)$ and $k_i = 2k_{i+1} = |G|/L$:

$$\frac{k_{i+1}}{|G|} \left(1 - \frac{k_i}{|G|}\right) = \frac{1}{2L} \left(1 - \frac{1}{L}\right) = \frac{L-1}{2L^2} \tag{25}$$

Thus:

$$\Pr(E_i) \geq 1 - \exp\left(-\frac{|G|(L-1)}{2L^2}\right) \tag{26}$$

By union bound:

$$\Pr\left(\bigcup_{i=1}^{L-1} E_i^c\right) \geq \max_i \Pr(E_i^c) = \Omega(1/L) \tag{27}$$

Therefore, random supervision leads to violations with constant probability, independent of training quality or sample size. The above probability multiplied by PAC bound gives the desired bound. $\square$

**Theorem 2.**

*Theorem* (Explicit Minimisation of Ordering Mismatches for Attributes). For an input $x$ with true label $y$, define:

$$\mathcal{A}_i(x) = \{a \mid \hat{a}_i^c(x) \geq \tau_a\}, \tag{28}$$

where $\tau_a \in [0, 1]$ is a threshold, typically chosen as $0.5$. Then, given $|\widehat{\ell_{\text{total}}} - \ell_{\text{total}}^*| \leq \epsilon$,

$$\Pr_{f^{\text{foca}}} (\mathcal{A}_i(x) \not\subseteq \mathcal{A}_{i+1}(x)) << \Pr_{f^{\text{rnd}}} (\mathcal{A}_i(x) \not\subseteq \mathcal{A}_{i+1}(x)). \tag{29}$$

*Proof.* The proof is similar to Theorem 1, except with the ordering changed. $\square$

## Proof of Theorem 4.2

**Theorem 4.2** (Information-Theoretic Benefit of FCA Supervision). *Consider a FoCA-CBM trained with formal concept lattice $\mathcal{L}$ constructed from $\langle G, M, I \rangle$. Let network layer $j$ be supervised by lattice level $i$ via class groups $G^i$ and attribute sets $M_i$. Then, under bounded training error $|\hat{\ell} - \ell^*| \leq \epsilon$ with $N$ training samples, the $\epsilon$-calibrated information gain of the network for layer $j$ is:*

$$I_{\mathcal{D}}(f_j(X); Y) - I_{\mathcal{D}}(f_{j-1}(X); Y) \geq \Delta_{\text{lattice}}^{(i)} - 2\Delta_{\text{align}}(\epsilon),$$

*where $\Delta_{\text{align}}(\epsilon) = O(\sqrt{\epsilon \log |G|} + N^{-1/2})$.*

**Notation.** For a sample $(x, y)$ let

$$\hat{s}_j[c] := s_j[c] \cdot \sigma(s_{j-1}[c]) \qquad \text{and} \qquad p_j[c] := \sigma(\hat{s}_j[c])$$

where $\sigma(z) = 1/(1 + e^{-z})$. Let $m := |G^i(y)|$. The group BCE loss at layer $j$ is

$$\ell_{\text{BCE},j}^{\text{group}}(x, y) = - \sum_{c \in G^{(i)}(y)} \log p_j[c] - \sum_{c \notin G^{(i)}(y)} \log(1 - p_j[c]).$$

**Assumption A3.6** (BCE Loss Bound on Groups). For layer $j$ supervised by lattice level $i$, the group-level BCE loss satisfies:

$$\mathbb{E}_{(x,y)\sim\mathcal{D}} \left[ \ell_{\text{BCE}_j}^{\text{group}} \right] = \mathbb{E}_{(x,y)} \left[ \sum_{c \in G} \mathbf{1}_{c \in G^{(i)}(y)}(- \log \sigma(\hat{s}_j[c])) + \mathbf{1}_{c \notin G^{(i)}(y)}(- \log(1 - \sigma(\hat{s}_j[c]))) \right] \leq \beta\epsilon, \qquad (30)$$

where $\beta > 0$ is the weight from Equation (3), and $\hat{s}_j = \sigma(s_j \cdot \sigma(s_{j-1}))$ is the refined prediction.

**Assumption A3.7** (Cluster-Density Matching). Layer $j$ is selected via the class-cluster density procedure such that:

$$|\text{ClassClusterDensity}(j) - \mathbb{E}_{\text{fc}\in\mathcal{L}_i}[\|\text{fc.extent}\|]| \leq \gamma, \qquad (31)$$

where $\gamma = O(N^{-1/2})$ from finite-sample concentration.

**Lemma 1 (Average sigmoid is close to 1 on the true group).** For any sample $(x, y)$ whose per-sample group loss satisfies $\ell_{\text{BCE},j}^{\text{group}}(x, y) \leq K\epsilon$ (which holds for most samples by Markov/Chebyshev from the expectation bound), we have

$$\frac{1}{m} \sum_{c \in G^{(i)}(y)} p_j[c] \geq \exp\left(-\frac{K\epsilon}{m}\right) = 1 - O\left(\frac{\epsilon}{m}\right). \qquad (32)$$

*Proof.* For the classes in the true group apply Jensen to $-\log$:

$$-\log\left(\frac{1}{m} \sum_{c \in G^{(i)}(y)} p_j[c]\right) \leq \frac{1}{m} \sum_{c \in G^{(i)}(y)} -\log p_j[c] \leq \frac{K\epsilon}{m}.$$

Exponentiating yields (32). $\square$

**Lemma 2 (Pigeonhole / concentration - most individual sigmoids are high).** Define $a := \dfrac{K\epsilon}{m}$. For any $\delta \in (0, 1)$ let $\alpha$ be the fraction of classes $c \in G^i(y)$ with $p_j[c] \leq 1 - \delta$. Then

$$\alpha \leq \frac{a}{\delta}.$$

Choosing $\delta = \sqrt{a}$ gives $\alpha \leq \sqrt{a} = O\left(\sqrt{\frac{\epsilon}{m}}\right)$. Hence at least a $1 - O(\sqrt{\epsilon/m})$ fraction of classes in the true group satisfy

$$p_j[c] \geq 1 - O\left(\sqrt{\frac{\epsilon}{m}}\right).$$

*Proof.* If $\alpha m$ classes have $p_j[c] \leq 1 - \delta$ then the group average is $\leq \alpha(1 - \delta) + (1 - \alpha) \cdot 1 = 1 - \alpha\delta$. Combine with (32) to get $\alpha\delta \leq a$, hence $\alpha \leq a/\delta$. $\square$

**Lemma 3 (From high sigmoid to large refined logit and per-class logit separation).** Let $\tau := \sqrt{\dfrac{K\epsilon}{m}}$ (so $\tau \to 0$ with $\epsilon \to 0$). For the majority of classes in $G^i(y)$ (fraction $1 - O(\tau)$) we have

$$p_j[c] = \sigma(\hat{s}_j[c]) \geq 1 - \tau \quad \Longrightarrow \quad \hat{s}_j[c] \geq \sigma^{-1}(1 - \tau) = \log\frac{1 - \tau}{\tau} =: L_j^+.$$

Since $\hat{s}_j[c] = s_j[c] \cdot \sigma(s_{j-1}[c])$ and (by earlier-level supervision) for classes that were kept at layer $j-1$ we have $\sigma(s_{j-1}[c]) \geq 1/2$ (for well-trained earlier layer), we deduce for those classes

$$s_j[c] \geq \frac{L_j^+}{\sigma(s_{j-1}[c])} \geq L_j^+.$$

Similarly, for most classes outside the true group one obtains $p_j[c] \leq \tau$ and hence $\hat{s}_j[c] \leq \sigma^{-1}(\tau) = -\log\frac{1-\tau}{\tau} =: L_j^- < 0$ and therefore $s_j[c] \leq L_j^-/\sigma(s_{j-1}[c]) \leq L_j^- < 0$. Thus the majority of in-group logits are $\geq L_j^+$ and the majority of out-group logits are $\leq L_j^-$, and

$$L_j^+ - L_j^- \;=\; \Theta(\log(1/\tau)) \;=\; \Theta\!\left(\tfrac{1}{2}\log\frac{m}{\epsilon}\right).$$

**Step 4 (Conditional entropy alignment).** Fix a typical sample $(x, y)$ for which the sigmoid-concentration conclusions hold, and let $\zeta$ denote the total probability mass (under the model's per-class probabilities implied by the refined sigmoids) on classes outside the true group $G^i(y)$. From Lemmas 1–3 we have $\zeta = O(\tau) = O\!\left(\sqrt{\frac{\epsilon}{m}}\right)$ (up to constants and averaging effects). The conditional entropy of $Y$ given the representation $f_j(x)$ satisfies

$$H(Y \mid f_j(x)) \leq (1 - \zeta)\log m + H_{\text{bin}}(\zeta) + \zeta\log(n),$$

where $H_{\text{bin}}(\zeta) = -\zeta\log\zeta - (1 - \zeta)\log(1 - \zeta) = O(\zeta\log(1/\zeta))$. Therefore

$$H(Y \mid f_j(x)) = \log m + O(\zeta\log n) = \log|G^{(i)}(y)| + O\!\left(\sqrt{\frac{\epsilon}{m}}\log n\right).$$

Taking expectation over $(x, y)$ and combining with finite-sample error from the class-cluster density matching (Assumption $\gamma = O(N^{-1/2})$) yields

$$\left| H_{\mathcal{D}}(Y \mid f_j(X)) - \mathbb{E}_y\big[\log|G^i(y)|\big] \right| \leq \Delta_{\text{align}}(\epsilon),$$

with

$$\Delta_{\text{align}}(\epsilon) = O\!\left(\sqrt{\epsilon\log n} + N^{-1/2}\right).$$

This is the claimed alignment error.

**Final step (Information-gain comparison).** Using $I(Y; f_j(X)) = H(Y) - H(Y \mid f_j(X))$ and the above alignment,

$$I_{\mathcal{D}}(f_j(X); Y) = H(Y) - \mathbb{E}_y[\log|G^i(y)|] \pm \Delta_{\text{align}}(\epsilon) = I_{\text{lattice}}^{(i)}(Y) \pm \Delta_{\text{align}}(\epsilon).$$

Similarly for layer $j-1$ aligned with level $i-1$ we get

$$I_{\mathcal{D}}(f_{j-1}(X); Y) = I_{\text{lattice}}^{(i-1)}(Y) \pm \Delta_{\text{align}}(\epsilon).$$

Subtracting,

$$I_{\mathcal{D}}(f_j(X); Y) - I_{\mathcal{D}}(f_{j-1}(X); Y) \geq \Delta_{\text{lattice}}^{(i)} - 2\Delta_{\text{align}}(\epsilon). \quad \square$$

## A4. More Analysis

**Impact of Class-Ordering:** We empirically validate Theorem 4.1 by training a FoCA CBM on ImageNet100 with random class ordering. Here, a sample is supervised by the right group with a probability of 0.5 and is supervised by a wrong group with a probabililty of 0.5. We observe that this leads to a sharp fall in test accuracy ($91.36 \to 28.96$), showing the importance of class ordering.

**Impact of CBM Training Mode:** There are several ways of learning concept-based models: *independent, sequential* and *joint* optimization. All the results in the paper are over *joint* models, following standard practice. To study the impact of this, we train FoCA CBMs using the other training schemes on the ImageNet100 dataset. Interestingly, we observe significant performance differences across training schemes: *Joint* - $91.88_{\pm 0.35}$, *Sequential* - $86.69_{\pm 0.04}$, *Independent* - $77.54_{\pm 0.35}$. We attribute this to two factors: (1) Broken iterative refinement: training attribute layers without class feedback breaks

Table A9. Effect of $\alpha$ on FoCA CBM on accuracy.

| Dataset | Alpha | FoCA CBM |
|---|---|---|
| AWA2 | 0.01 | 92.36 |
| | 0.1 | 92.13 |
| | 1 | 91.94 |
| CIFAR100 | 0.01 | 79.36 |
| | 0.1 | 75.84 |
| | 1 | 73.50 |
| Imagenet100 | 0.01 | 91.92 |
| | 0.1 | 87.73 |
| | 1 | 87.56 |

Table A10. Effect of $\beta$ on FoCA CBM on accuracy.

| Dataset | Beta | FoCA CBM |
|---|---|---|
| AWA2 | 0.01 | 92.36 |
| | 0.1 | 91.24 |
| | 1 | 90.02 |
| CIFAR100 | 0.01 | 79.36 |
| | 0.1 | 71.85 |
| | 1 | 65.09 |
| Imagenet100 | 0.01 | 91.92 |
| | 0.1 | 86.94 |
| | 1 | 85.22 |

the coordination our cascading mechanism relies on; (2) Compounding error: errors at earlier layers cascade through the hierarchy. This implies that *joint* optimization is essential, and high attribute accuracy in isolation is insufficient.

**Choice of $\alpha$ and $\beta$:** We perform ablation studies on the two hyperparameters $\alpha$ and $\beta$. We fix one of the values of these hyperparameters from the best models and vary the other one. Both are varied among [0.01, 0.1, 1]. The results are reported in Tables A9 and A10. A lower value is preferred since the first two terms of our loss function in Eqn 4 are summed up over multiple layers, thus scaling the loss values. However, a more complete picture is obtained by also examining the other aspect we are concerned about, namely the goodness of the structure of the embeddings, which we study over the ImageNet100 dataset. Since our overall loss is a multi-objective optimization problem, there is naturally a trade off between accuracy and structure. As we can see from the below tables, giving more weight to the class grouping term leads to more coherent cluster groups (as indicated by the lower CI and DBI values) but at the cost of accuracy. Similarly, increasing the weight of the attribute loss leads to higher attribute accuracy, but degrades accuracy and clustering quality. This highlights the critical role that the auxiliary losses play in the learning. Crucially, setting either $\alpha = 0$ or $\beta = 0$ leads to dramatic accuracy drops (30.07 and 42.91 respectively vs. 91.92 at $\alpha=\beta=0.01$); CI and DBI are also poor due to the lack of proper supervision, confirming both loss terms are essential.

| $\beta$ | Acc | CI | DBI |
|---|---|---|---|
| 0.01 | **91.92** | 0.573 | 1.862 |
| 0.1 | 86.94 | 0.566 | 1.816 |
| 1 | 85.22 | **0.549** | **1.712** |

| $\alpha$ | Acc | CI | DBI | Attr Acc |
|---|---|---|---|---|
| 0.01 | **91.92** | **0.573** | **1.862** | 88.48 |
| 0.1 | 87.73 | 0.6218 | 2.006 | 92.55 |
| 1 | 88.6 | 0.656 | 2.024 | **95.21** |

| $(\alpha, \beta)$ | Acc | CI | DBI |
|---|---|---|---|
| $(0, 0.01)$ | 30.07 | 0.675 | 2.257 |
| $(0.01, 0)$ | 42.91 | 0.641 | 2.075 |
| $(0, 0)$ | 29.24 | 0.6835 | 2.3902 |

Overall, these results suggest that the auxiliary loss terms are most effective when used as light regularizers, striking a balance between accuracy and semantic structure, which is consistent with standard multi-objective learning setups.

**More Cluster-based Analysis:** We provide more comprehensive results on our cluster-based analysis here. Going beyond the reported average *CI* and *DBI* scores in Table 1, we examine herein how these metrics evolve across blocks of a model. Ideally, both scores should decrease as we move deeper into a network, indicating lower cluster impurity and more compact, well-separated representations. In other words, we would want embeddings to move closer to the origin of the CI-DBI space across blocks. As is evident from Figure A7 for CIFAR-100, the FoCA CBM models move closest to the origin. In contrast, some baselines show little change across layers, suggesting weaker representation refinement. The *CI-DBI* plots for the other datasets are also provided. We also use line plots to visualize cluster impurity reduction alone through blocks (Fig A11) and see that our models gradually reduce in cluster impurity, unlike the other models where there is either a sharp drop in the last block or no drop at all. This indicates our models' ability to learn more meaningful embeddings. Finally, we provide more comprehensive t-SNE plots comparing the cluster formation in all blocks of the respective backbone networks in Fig A13, A14, A15. The set of classes considered for AwA2 were *dalmatian, german shepherd, horse*, for ImageNet100 were *agama, banded-gecko, whiptail* and for CIFAR100 were *beetle, butterfly, cockroach*.

**More Results on ViT-based backbones:** We obtained results on more baselines with ViT backbones on CIFAR100 and see that our method continues to outperform baselines. The nature of the representations of ViTs is less explicitly understood than CNNs. One could view our method as providing an explicit inductive bias for hierarchical representations on such

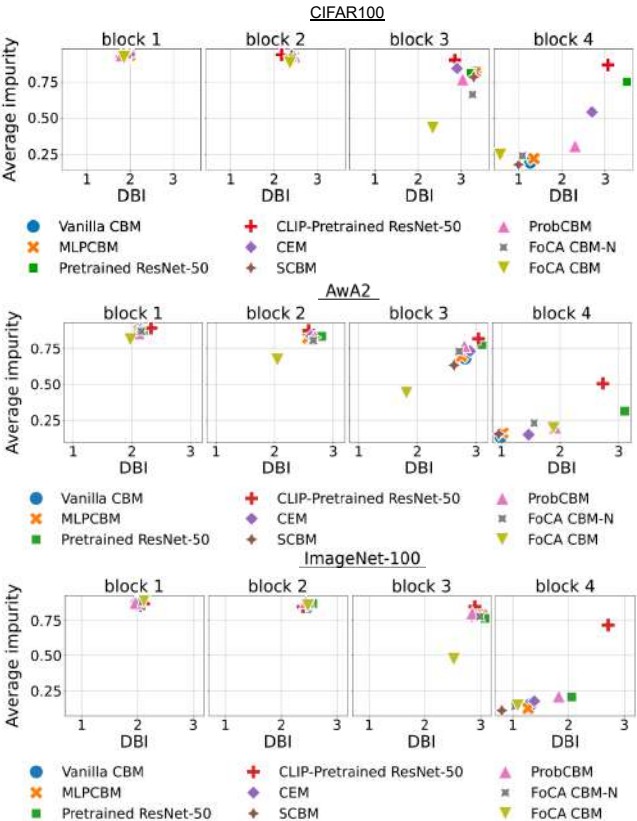

*Figure A7.* CI vs DBI plot per block on CIFAR100, AwA2 and ImageNet100 datasets. Each marker indicates a model trained on the respective dataset. Markers should ideally move towards the origin in higher blocks. We attribute the superiority of CBMs and MLPCBMs on AwA2 to the simplicity of the dataset.

architecture. Recent works (Park et al., 2025) show that hierarchical structure can emerge in the representation space of transformers. Our results provide empirical evidence that FoCA CBM's inductive bias successfully uncovers this structure in ViTs, as reflected by consistent gains across all three metrics.

## A5. Dataset Details

- **AwA2:** The Animals with Attributes (AwA2) dataset (Xian et al., 2019) is commonly used for zero-shot learning (ZSL) and attribute-based classification. It consists of 37322 images (26125 training, 11197 testing) of 50 animal classes, annotated with 85 numeric attribute values for each class and is class-level expert annotated. Each class in the dataset has on average $31 \pm 4$ active attributes, with 22 being the minimum number of attributes active for a particular class and 39 being maximum.

*Table A11.* Comparison of baselines with ViT-backbones. FoCA ViT outperforms over all metrics.

| ViT-Based Model | Acc | CI | DBI |
|---|---|---|---|
| ViT CBM | $84.49 \pm 0.55$ | $0.774 \pm 0.016$ | $2.237 \pm 0.006$ |
| LFCBM | $72.40 \pm 0.27$ | $0.873 \pm 0.000$ | $2.797 \pm 0.000$ |
| Posthoc CBM | $71.29 \pm 0.63$ | $0.873 \pm 0.000$ | $2.797 \pm 0.000$ |
| LaBo | $76.12 \pm 1.04$ | $0.873 \pm 0.000$ | $2.797 \pm 0.000$ |
| CFCBM | $74.11 \pm 0.44$ | $0.873 \pm 0.000$ | $2.797 \pm 0.000$ |
| FoCA ViT | $\mathbf{86.65 \pm 0.30}$ | $\mathbf{0.755 \pm 0.004}$ | $\mathbf{1.983 \pm 0.021}$ |

*Table A12.* Details about the formal concept lattice obtained for each dataset.

| Dataset | $|G|$ | $|M|$ | Fill ratio | $|\mathcal{L}|$ | $L$ | Worst case $|\mathcal{L}|$ | Time (s) | Space (MB) |
|---|---|---|---|---|---|---|---|---|
| AwA2 | 50 | 85 | 0.368 | 64315 | 26 | $1.13 \times 10^{15}$ | 37.37 | 63.62 |
| CIFAR100 | 100 | 700 | 0.023 | 915 | 10 | $1.27 \times 10^{30}$ | 1.97 | 1.00 |
| ImageNet-100 | 100 | 700 | 0.026 | 1593 | 10 | $1.27 \times 10^{30}$ | 4.56 | 1.68 |

- **CIFAR100:** CIFAR100 is a well-known subset of the Tiny Images dataset (Krizhevsky et al., 2009) comprising 100 classes. Attributes associated with each of these classes are generated using GPT-3 (Oikarinen et al., 2023), where an initial concept set is generated through queries like "List the most important features for recognizing something as a {class}". This is followed by a sequence of filtering steps, to improve the quality of the concept set. This involves the $k$-means clustering (with $k = 700$) of similar attributes together based on their `all-MiniLM-L6-v2` embeddings and choosing the closest attributes to each cluster centroid as the cluster representatives. The dataset finally consists of 60000 images (50000 for training and 10000 for testing), 700 attributes and 100 classes. Each class in the dataset has on average $16 \pm 3$ active attributes, with 8 being the minimum number of attributes active for a particular class and 24 being maximum.

- **ImageNet100:** ImageNet100 is a well-known subset of the ImageNet-1k dataset (Russakovsky et al., 2015). We randomly select 100 classes from the set of 1k classes. We follow the same process we followed for the CIFAR100 dataset here as well and acquire attributes from GPT-3. The dataset finally consists of 134973 images (129973 training, 5000 testing) of 100 distinct classes and 700 LLM-generated unique attributes. Each class in the dataset has on average $18 \pm 3$ active attributes, with 9 being the minimum number of attributes active for a certain class and 25 being maximum.

Some example classes and their corresponding attributes are provided in Tab A13.

## A6. Lattice Details

Our formal concept lattices are constructed using the `concepts` [1] Python module. This module employs the Fast Concept Analysis algorithm (Troy et al., 2007) for generating the lattices (CONSTRUCTLATTICE in Algorithm 1). The hierarchy level of each formal concept is computed by first performing a topological sort on the lattice (which is always a directed acyclic graph), and then iteratively updating the level of the upper neighbors of each formal concept traversed in topological order (described in Algorithm 2).

- The **AwA2** lattice consists of 50 classes, 85 attributes, and 64315 total formal concepts across 26 hierarchy levels with 1, 50, 743, 3038, 5755, 7440, 7876, 7472, 6680, 5738, 4800, 3912, 3083, 2310, 1693, 1221, 873, 613, 409, 262, 165, 98, 52, 23, 7, 1 formal concepts per level. Computing the lattice took $37.37$ seconds on average, and it occupies $63.62$ MB of space.

- The **CIFAR100** lattice consists of 100 classes, 700 attributes and 915 total formal concepts across 10 hierarchy levels with 1, 100, 345, 211, 131, 75, 33, 14, 4, 1 formal concepts per level. Computing the lattice took $1.97$ seconds on average, and it occupies $1.00$ MB of space.

- The **ImageNet-100** lattice consists of 100 classes, 700 attributes, and 1593 total formal concepts across 10 hierarchy levels. The number of formal concepts in the 10 levels is 1, 100, 592, 415, 250, 129, 65, 30, 10, 1, respectively, going from the infimum to the supremum. Computing the lattice took $4.56$ seconds on average, and it occupies $1.68$ MB of space.

## A7. Implementation Details

The algorithm for the whole method is provided in Algorithm 1.

---

[1] https://pypi.org/project/concepts/

**Algorithm 2** COMPUTE HIERARCHY LEVELS($\mathcal{L}$):

**Require:** Formal concept lattice $\mathcal{L}$.
1: $\mathcal{L}_s \leftarrow$ TopologicalSort($\mathcal{L}$)                    ▷ Infimum at the first index
2: level[fc] $\leftarrow 0 \quad \forall$ fc $\in \mathcal{L}_s$
3: **for** $u \in \mathcal{L}_s \setminus \{\mathcal{L}_s.\text{infimum}\}$ **do**
4:     **for** $v$ in $u.\text{upper\_neighbors}$ **do**
5:         level[$v$] = max{level[$v$], level[$u$] + 1}
6:     **end for**
7: **end for**
8: **return** level

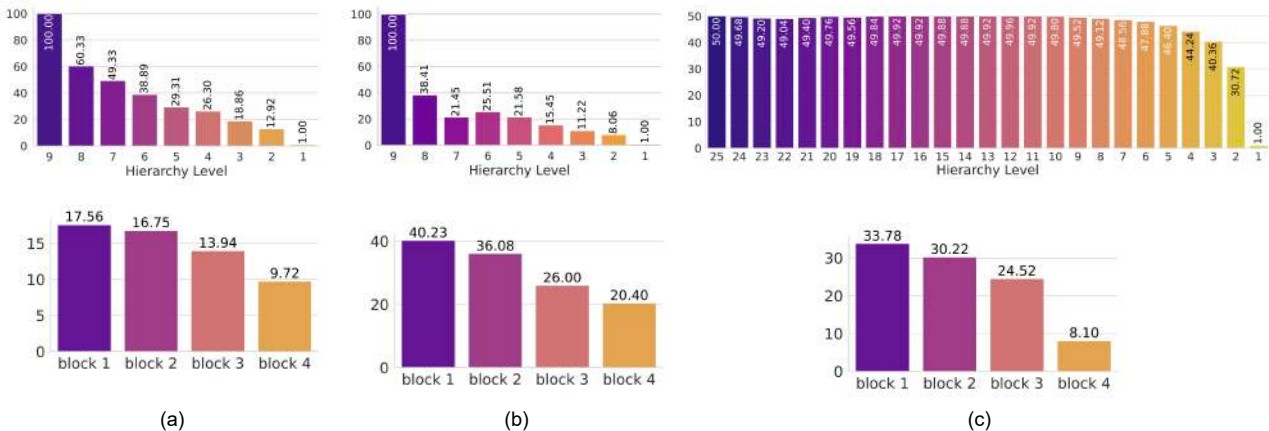

*Figure A8.* The average number of classes active per lattice level (top) and per block of a ResNet (bottom) for the ImageNet-100 (a), CIFAR100 (b) and AwA2 (c) datasets.

**Evaluation Metric Details:** We provide additional details herein on the clustering-based metrics (*CI, DBI*) we report in our results in the main paper (Table 1). At each block, we get the set of embeddings on the samples for all $n$ classes and cluster them (with $k = n$). For each cluster in a block, we compute the `gini-index` and `davies_bouldin_score`. Averaging this value over all the clusters in a block, gives the average impurity and average cluster compactness of that block. We then take the harmonic mean of this number across all blocks to get the numbers representing the average *CI* and *DBI* of the model. These are described in Algorithm 3 and 4.

**Model Details:** All models were run on a single NVIDIA GeForce RTX 3090. We use a ResNet18-based backbone for all AwA2 models and a ResNet50-based backbone for all CIFAR100 and ImageNet100 models. On our models, we place semantic layers at the end of blocks and hence have 4 backbone position choices corresponding to the 4 blocks in a ResNet. All the results reported in the main table are models with 2 semantic layers.

**Algorithm 5** CLASSCLUSTERDENSITYMODEL($f, \mathcal{D}$):

**Require:** Model $f$, Dataset $\mathcal{D}$.
  Initialize avg\_class\_per\_block $\leftarrow \mathbf{0}$
  $n \leftarrow$ number of classes in $\mathcal{D}$
  **for** b in $f$.blocks **do**
      **for** j in $\mathcal{D}$ **do**
          $E_{b_i} = E_{b_i} \cup \{h_i^j\}$                    ▷ Set of embeddings at block $b_i$
      **end for**
      clusters = `k-means`($E_{b_i}, n$)                    ▷ Cluster $E_{b_i}$ with $n$ centers
      **for** c in clusters **do**
          avg\_class\_per\_block[b] += `UniqueClasses`(c) ▷ gets number of unique classes, associated with the embeddings, in a cluster
c
      **end for**
      avg\_class\_per\_block[b] /= $n$
  **end for**
  **return** avg\_class\_per\_block

---

**Algorithm 3** COMPUTE CLUSTER IMPURITY($f, \mathcal{D}$):

---

**Require:** Model $f$, Dataset $\mathcal{D}$.
1: **Initialize** ci = 0
2: **for** b in $f$.blocks **do**
3:     **for** j in $\mathcal{D}$ **do**
4:         $E_{b_i} = E_{b_i} \cup \{h_i^j\}$                                   ▷ Set of embeddings at block $b_i$
5:     **end for**
6:     clusters = k-means($E_{b_i}, n$)                              ▷ Cluster $E_{b_i}$ with $n$ centers
7:     **for** c in clusters **do**
8:         ci += gini-index(c)
9:     **end for**
10:     ci /= $n$
11: **end for**
12: ci /= #blocks
13: **return** ci

---

**Algorithm 4** COMPUTE DBI($f, \mathcal{D}$):

---

**Require:** Model $f$, Dataset $\mathcal{D}$.
**Initialize** dbi = 0
**for** b in $f$.blocks **do**
    **for** j in $\mathcal{D}$ **do**
        $E_{b_i} = E_{b_i} \cup \{h_i^j\}$                                   ▷ Set of embeddings at block $b_i$
    **end for**
    clusters = k-means($E_{b_i}, n$)                              ▷ Cluster $E_{b_i}$ with $n$ centers
    dbi += davies_bouldin_score($E_{b_i}$, clusters.labels)
**end for**
dbi /= #blocks
**return** dbi

---

**Algorithm 6** CLASSCLUSTERDENSITYLATTICE($\mathcal{L}$):

---

**Require:** Lattice $\mathcal{L}$
Initialize avg_class_per_level $\leftarrow$ **0**
**for** level in $\mathcal{L}$ **do**
    count $\leftarrow$ 0
    **for** fc in level **do**
        count += 1
        avg_class_per_level[level] += len(fc.extent)
    **end for**
    avg_class_per_level[level] /= count
**end for**
**return** avg_class_per_level

---

**Computational Complexity:** Our FoCA CBM models were trained in $\approx$: 40 min for AwA2, 1.75 hrs for CIFAR100 and 3 hrs for ImageNet100 on a single NVIDIA GeForce RTX 3090; these times were about the same timings as Vanilla CBMs and MLPCBMs took. Our lattices are generated offline before training and took $\approx$: 37 secs for AwA2, 2 secs for CIFAR100 and 4.5 secs for ImageNet100, thus making this a near-negligible cost.

**Hyperparameter Details:**

- *Vanilla CBMs and MLPCBMs*: All models here were trained for 30 epochs with a batch size of 128, an AdamW optimizer and a onecycle scheduler. The AwA2 and CIFAR100 models were trained using a learning rate of $3 \times 10^{-4}$, while the ImageNet100 models were trained with a learning rate of $1 \times 10^{-4}$.

- *Posthoc CBMs*: Here, we use the multimodal CLIP-based backbones for AwA2 and ImageNet100 datasets, with $\lambda = 2 \times 10^{-4}$ and a batch size of 512. For CIFAR100, we take the numbers from the respective paper (Posthoc CBMs).

- *Label-Free CBMs*: Most hyperparameters used here were the same as the ones reported by the paper on the ImageNet dataset. Additionally, we use a clip-cutoff of 0.26 for ImageNet100 models and 0.25 for AwA2 models. An Adam

---

**Algorithm 7** SELECTLAYERSANDLEVELS($f, \mathcal{D}, \mathcal{L}, m$)

---

**Require:** Model $f$, Dataset $\mathcal{D}$, Lattice $\mathcal{L}$, number of (layer,level) pairs to select $m \geq 1$
1: block_density $\leftarrow$ CLASSCLUSTERDENSITYMODEL($f, \mathcal{D}$)          $\triangleright$ length = #blocks = $L_f$
2: level_density $\leftarrow$ CLASSCLUSTERDENSITYLATTICE($\mathcal{L}$)          $\triangleright$ length = #levels = $L_{\mathcal{L}}$
3: $L_f \leftarrow \text{len}(block\_density)$     $L_{\mathcal{L}} \leftarrow \text{len}(level\_density)$
4: selected_layers $\leftarrow \left[\, L_f - 1 \,\right]$          $\triangleright$ index of last (top) block
5: selected_levels $\leftarrow \left[\, L_{\mathcal{L}} - 1 \,\right]$          $\triangleright$ index of most fine-grained lattice level
6: remaining $\leftarrow m - 1$
7: last_level_idx $\leftarrow L_{\mathcal{L}} - 1$          $\triangleright$ we will search levels strictly above this index
8: **for** layer_idx in $\text{range}(L_f - 2, -1, -1)$ **do**          $\triangleright$ iterate remaining model blocks bottom→top
9:      **if** remaining == 0 **then**
10:          **break**
11:      **end if**
12:      layer_density $\leftarrow$ block_density[layer_idx]
13:      chosen_level $\leftarrow$ None          $\triangleright$ search lattice levels from (last_level_idx - 1) downward to 0
14:      **for** level_idx in $\text{range}(\text{last\_level\_idx} - 1, -1, -1)$ **do**
15:          **if** level_density[level_idx] $\geq$ layer_density **then**
16:              chosen_level $\leftarrow$ level_idx
17:              **break**
18:          **end if**
19:      **end for**
20:      **if** chosen_level is not None **then**
21:          selected_layers.append(layer_idx)
22:          selected_levels.append(chosen_level)
23:          last_level_idx $\leftarrow$ chosen_level
24:          remaining $\leftarrow$ remaining $- 1$
25:      **end if**
26: **end for**
27: **return** selected_layers, selected_levels

---

optimizer with a learning rate of $1 \times 10^{-3}$ was used to learn the concepts. Finally, for learning the classes, we use glmsaga with a regularization strength of $1 \times 10^{-4}$ for ImageNet100 and $3 \times 10^{-4}$ for AwA2. We use a batch size of 512. We report the CIFAR100 results from the paper.

- *Concept Embedding Models*: We train all these models for 100 epochs with a batch size of 256, a learning rate of 0.01 and an SGD optimizer.

- *Language in a Bottle (LaBo)*: Since these models work with CLIP-based backbones, we use a CLIP:RN50, along with submodular concept selection, max epochs of 10000, batch size of 512 and learning rate of $1 \times 10^{-5}$ on all datasets.

- *Stochastic CBMs*: All models were trained for 70 epochs, with a batch size of 64, learning rate of $3 \times 10^{-5}$ using an Adam optimizer, with the number of monte carlo samples being 100.

- *Probabilistic CBMs*: All models were trained with an embedding size of 16, training intervention probability of 0.25, learning rate of $1 \times 10^{-2}$ with an SGD optimizer, a batch size of 256 and max epochs of 100.

- *Coarse-to-Fine CBMs*: For all the models here we first compute the CLIP similarities using a CLIP:RN50 model and then train the models. We use a learning rate of $3 \times 10^{-4}$ with an Adam optimizer, batch size of 256 and number of epochs of 30.

- *Hybrid CBMs*: Since these models work with CLIP-based backbones, we use a CLIP:RN50 on all datasets, along with submodular concept selection with a dynamic concept ratio of 0.5, max epochs of 5000 and learning rate of $5 \times 10^{-5}$. For the AwA2 and CIFAR100 models, we use a batch size of 512 and for the Imagenet100 models, we use a batch size of 4096.

- *FoCA CBMs*: Our hyperparameters here are the same as Vanilla CBM models.

**GPT-Hierarchy Details:** The prompt used for GPT4 to generate the LLM-Based hierarchy was the following: *Given a set of classes and attributes, generate 2 sets: a set of general attributes and a set of specific attribute, with the set of specific*

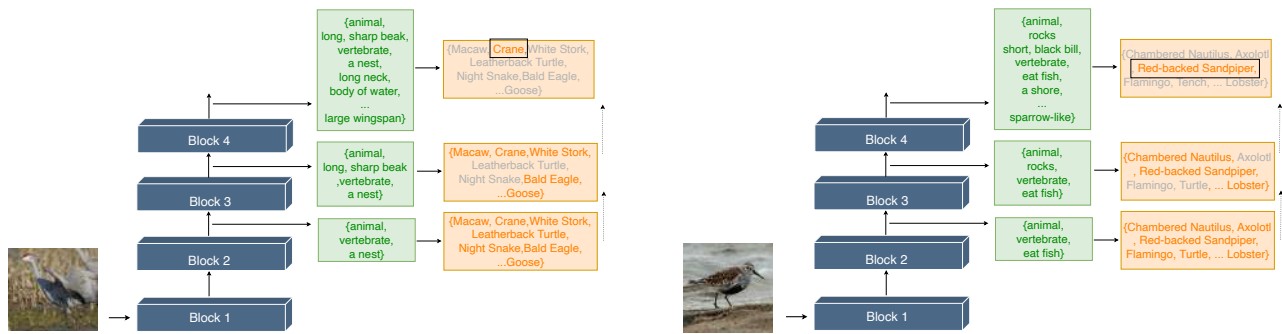

*Figure A9.* Some qualitative results of the predicted attributes and subsequent class group refinement for some samples from ImageNet100.

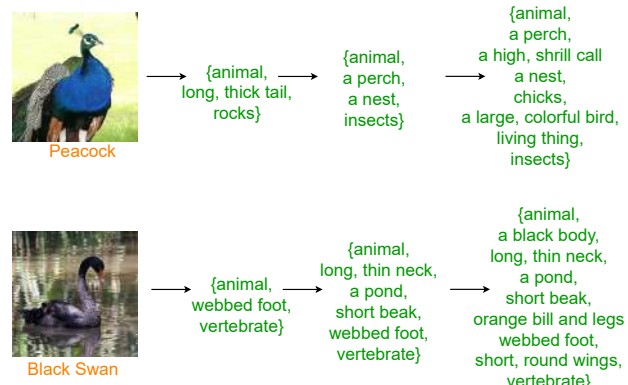

*Figure A10.* More examples of the attributes learned by a FoCA CBM after blocks 2, 3 and 4 of a ResNet50 for the classes *Peacock* and *Black Swan* in ImageNet100.

*attributes being a superset of general attributes. I also need a class group set that would get activated using the general attributes per class. For example: a general set of attributes could be "animal", "vertebrate", "mammal", "strong" and a specific set of attributes could be "animal", "vertebrate", "a long beak", "large wings", "mammal", "rocks", "strong" and for the class "macaw", a class group could be "indigo bunting, "macaw", "flamingo" which could be activated by a subset of the general attributes.*

This generates a two-level hierarchy which is compared with a two-level FoCA CBM.

## A8. Limitations

To facilitate future work, we also outline a few limitations of our approach. Firstly, as with all concept-based methods, the quality of our intermediate semantic representations is dependent on the accuracy of the attribute annotations. This dependency is particularly pronounced in LLM-annotated datasets such as CIFAR100 and ImageNet100. Enhancing annotation quality could therefore lead to notable gains in both model performance and interpretability. Secondly, once again mirroring a concern that is common across concept-based models, constraining the model to operate through semantic concepts can, in some cases, limit overall performance, a trade-off that is reflected in parts of our results. While we demonstrate consistent improvements in interpretability and related metrics, narrowing this performance gap remains a key area for further investigation. Finally, concept-based models typically treat concepts as static entities. Developing mechanisms that allow concept representations to adapt dynamically to the context of a specific input (e.g., a given input image) could be an interesting approach to improvements in flexibility and model performance.

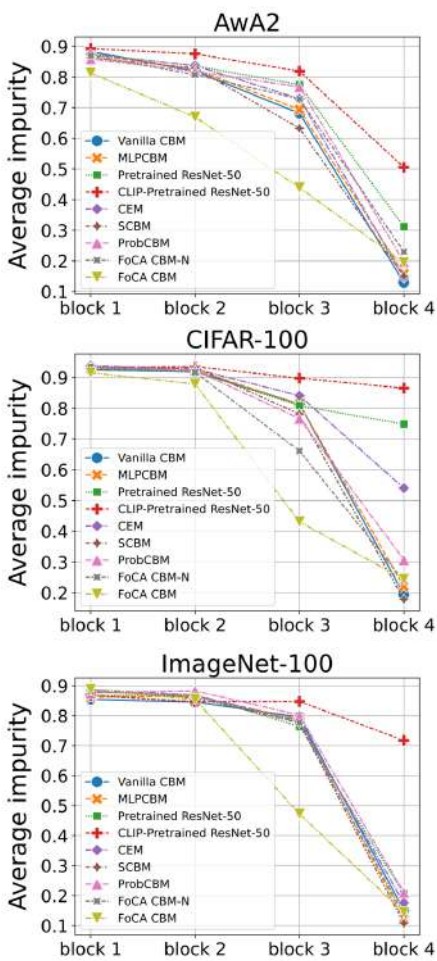

*Figure A11.* Cluster impurity over all models per block on AwA2, CIFAR100 and ImageNet100 datasets. Our models (inverted light green triangle) display a gradual reduction in impurity. The other models fall sharply at the last block or not at all.

*Table A13.* Examples of classes and subsets of corresponding attributes from AWA2, CIFAR100 and ImageNet100 datasets.

| Dataset | Class | Concepts |
|---|---|---|
| AwA2 | Raccoon | black, white, gray, patches, spots, stripes, furry, small, pads, paws, tail, chewteeth, meatteeth, claws, walks, fast, quadrapedal, active, nocturnal, hibernate, agility |
| | Cow | black, white, brown, patches, spots, furry, toughskin, big, bulbous, hooves, tail, chewteeth, horns, smelly, walks, slow, strong, quadrapedal, active, inactive |
| | Dolphin | white, blue, gray, hairless, toughskin, big, lean, flippers, tail, chewteeth, swims, fast, strong, muscle, active, agility, fish, newworld, oldworld, coastal, ocean, water |
| CIFAR100 | Chair | furniture, a person, object, legs to support the seat, an office, a computer, a desk, four legs, a backrest, armrests on either side |
| | House | windows, building, object, structure, a yard, a chimney, a door, a wall, siding or brick exterior, a garage, roof |
| | Kangaroo | a grassland, short front legs, an animal, a safari, mammal, a long, powerful tail, brown or gray fur, marsupial, long, powerful hind legs, Australia |
| ImageNet100 | Electric Ray | paddle-like fins, a flat circular shape, fish, a long, thick tail, mammal, animal, water, vertebrate, a large mouth, a large, bulky body |
| | White Stork | an animal, a large size, a tree, a field, insects, a sky, a long, curved neck, white feather, a thin neck, long red legs, long, arms and legs, a medium-sized body, vertebrate, a long orange beak |
| | Komodo Dragon | a large size, a keeper, scales, a tree, a dish, scaly skin, a rock, long, sharp claws, a long, thick tail, a long, forked tongue, an animal, reptile, a fence, vertebrate, a water dish, a zoo, a heat lamp, a large, bulky body, a cage, a lizard |

**ResNet**

**ViT**

*Figure A12.* More examples of the kind of attributes intervened on in the last layers versus an intermediate layer (chosen according to the severity of the misclassification). Intermediate layers have more general attributes.

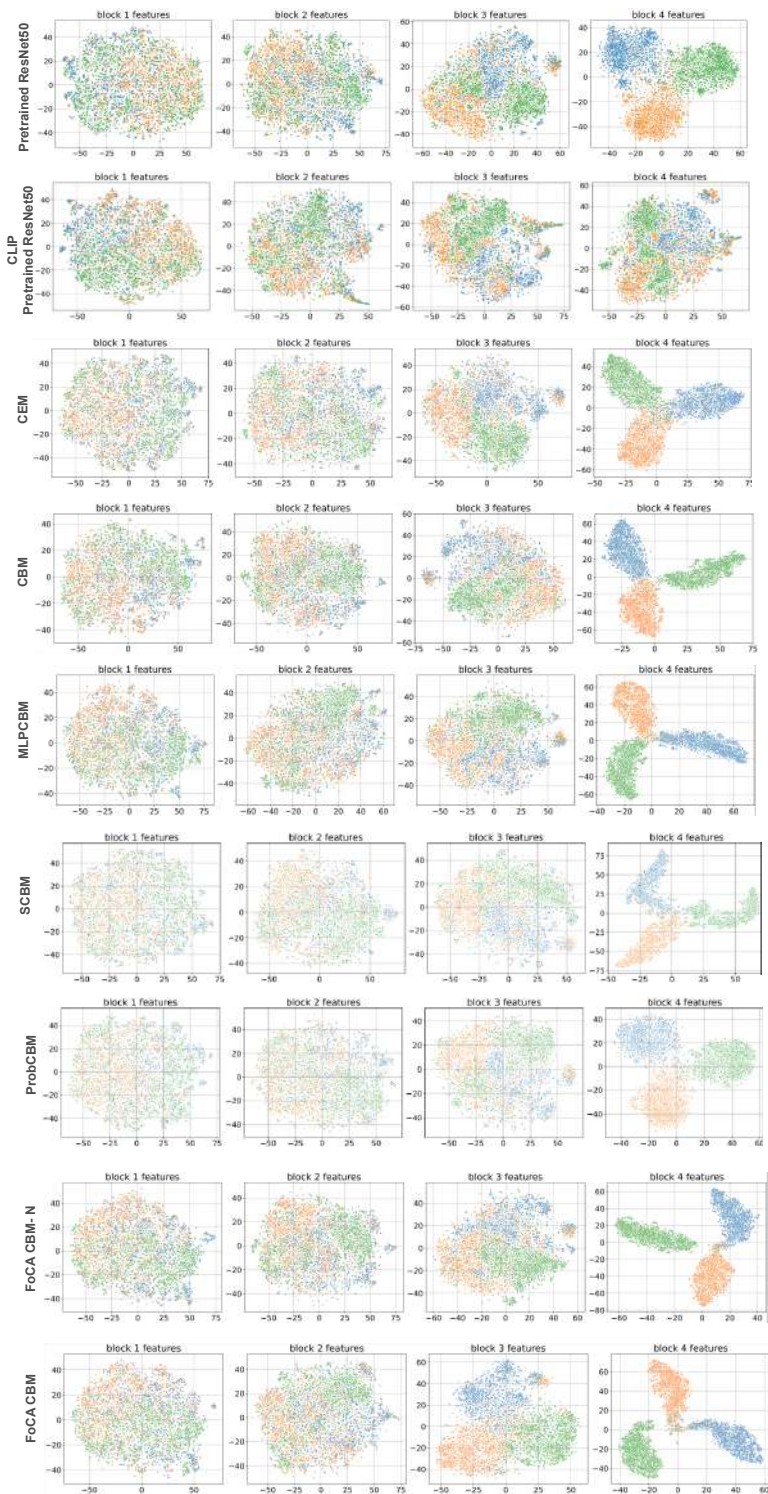

*Figure A13.* t-SNE plots of the embeddings obtained from the backbones of the models mentioned on the left of each plot on ImageNet100. On most models the clusters separate out only at the final block; in FoCA CBM, it happens gradually over blocks.

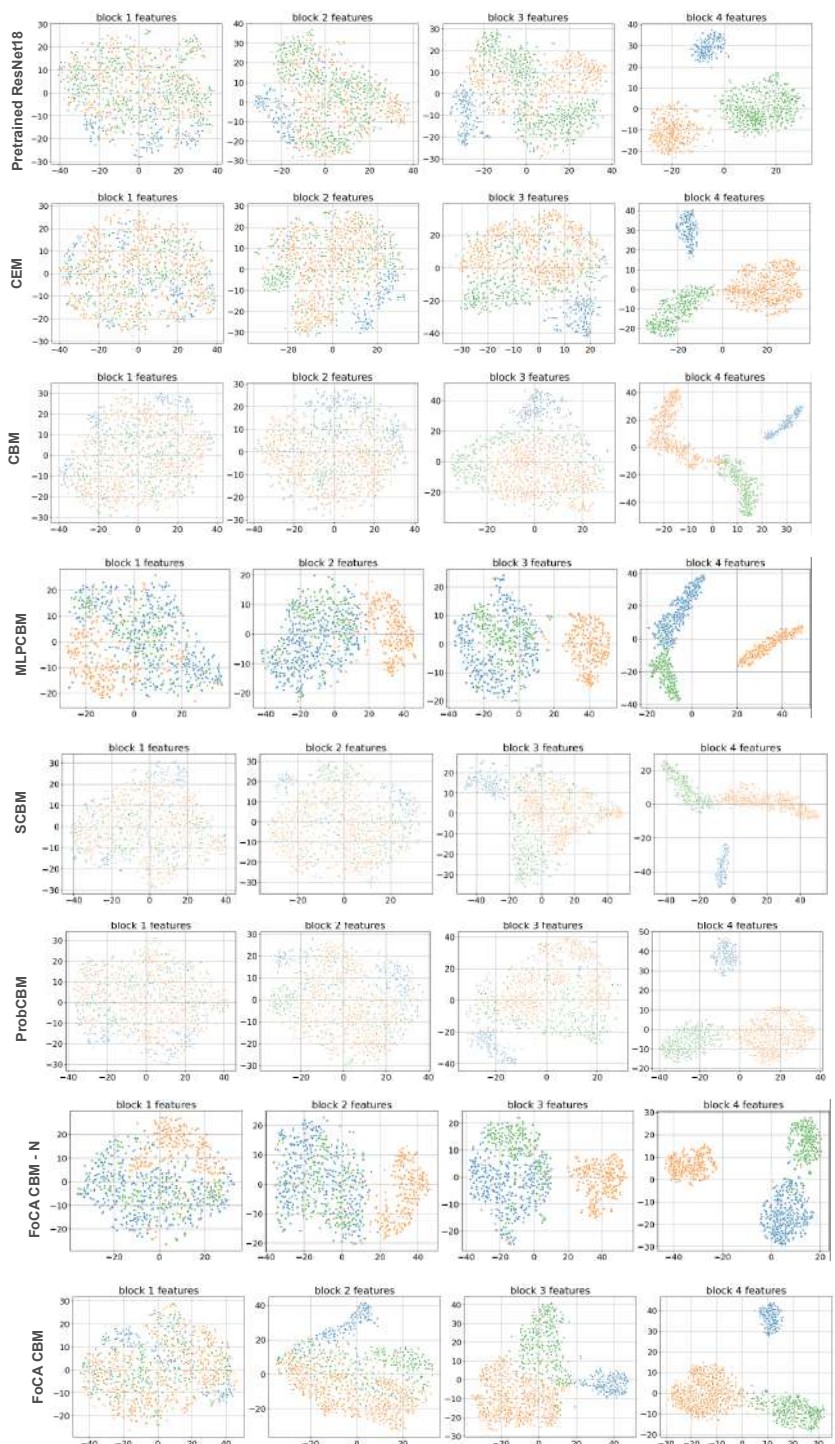

*Figure A14.* t-SNE plots of the embeddings obtained from the backbones of the models mentioned on the left of each plot on AwA2. On most models the clusters separate out only at the final block; in FoCA CBM, it happens gradually over blocks.

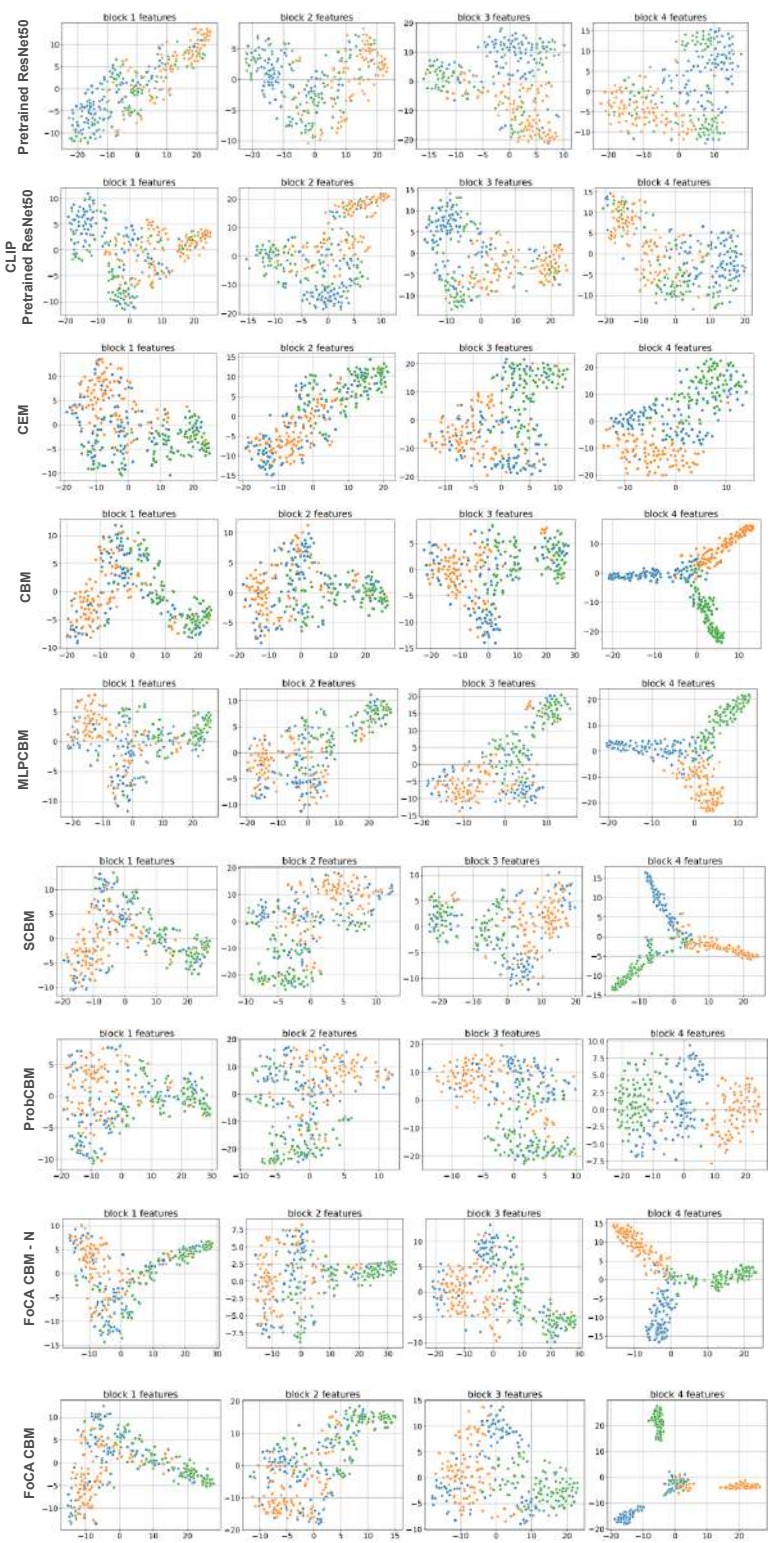

*Figure A15.* t-SNE plots of the embeddings obtained from the backbones of the models mentioned on the left of each plot on CIFAR100.

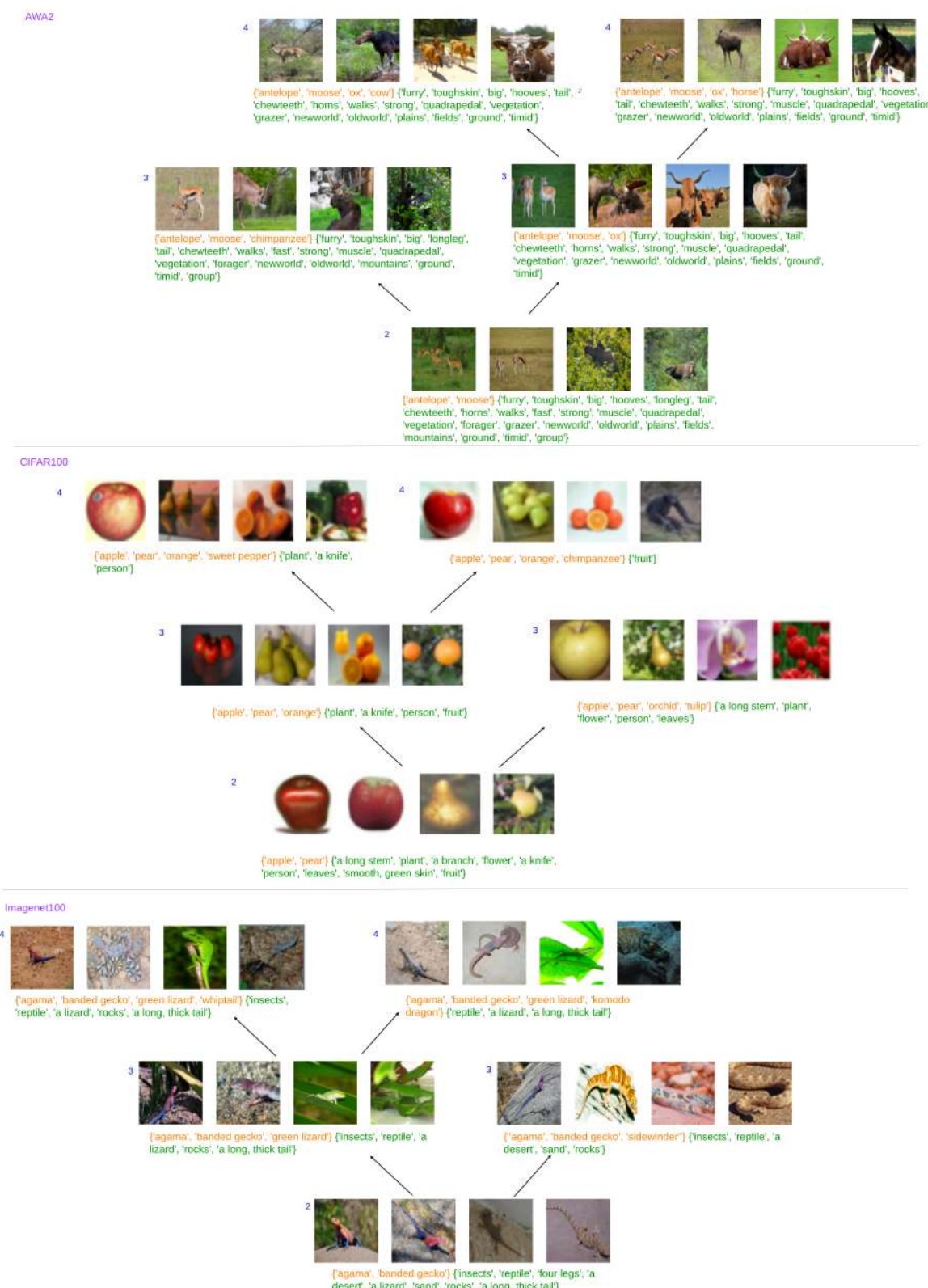

*Figure A16.* More examples of formal concepts (extent (classes) - intent (attributes)) from the lattices built for the AwA2, CIFAR100 and ImageNet100 datasets. The shown formal concepts have parent-children relations denoted by the arrows. The level that the formal concept belongs to is provided at the top left of each formal concept.

