# OpenReview forum: "Formal Concept Lattices are Good Semantic Scaffolds for Concept-Based Learning"
_ICML.cc/2026/Conference — ICML 2026 regular_

### Official Review · Reviewer_8xYn · 2026-03-10

**Soundness:** 3
**Presentation:** 3
**Significance:** 3
**Originality:** 3
**Overall Recommendation:** 4
**Confidence:** 3

**Summary:**

This paper proposes a formal concept analysis (FCA)-based concept-based model,
where a formal concept lattice provides a semantic hierarchy to guide different network layers toward learning coarse-to-fine concepts, rather than learning all concepts at a single flat bottleneck layer. The authors carefully design the framework to align concept lattice levels with layer depths using class-cluster density, and introduce layer-wise classifiers and iterative refinement to provide structured supervision across different semantic granularities. Extensive experiments and analyses on ImageNet100, AwA2, and CIFAR100 validates that the proposed method outperforms the baselines in terms of classification accuracy and semantic quality measured by clustering-based metrics.

**Compliance With Llm Reviewing Policy:**

Affirmed.

**Final Justification:**

The authors addressed all of the concern during the rebuttal, and I have no remaining concern.
The motivation for leveraging semantic hierarchies in concept bottleneck models is convincing and it is well formalized with formal concept lattices. Therefore, I decided to maintain my score.

**Key Questions For Authors:**

- Related to the first weakness in the strength&weakness section, could the authors clarify why the method remains effective even when the induced formal concept lattices are somewhat noisy?

- Although AwA2 uses expert annotations for attribute labels, which would be likely to be more reliable compared to GPT-based annotations used for ImageNet100 or CIFAR100, Table 2 shows that the concept leakage seems severe (only ~0.5% drop). Could the authors clarify why leakage remains substantial in this setting?

- Is Eq. (1) correct? The description in L137-140 is clear but Eq. (1) seems unclear. Shouldn't it be like
$A^\uparrow = \\{m \in M | \forall g \in A : (g, m) \in I \\}$ rather than $\forall g \in A : (x,y) \in I$, since x and y are not defined here.

**Limitations:**

Yes

**Strengths And Weaknesses:**

**Strengths**
- The paper is well organized and easy to follow.
- The motivation for leveraging semantic hierarchies in concept bottleneck models is convincing, and the idea is well formalized using formal concept lattices.
- The experimental results show that the proposed method outperforms the baselines and validate the improved interpretability of the embeddings.
- The empirical study is extensive and includes several insightful analyses that support the paper’s main hypothesis.


**Weaknesses**
- While the high-level idea of leveraging the semantic hierarchies in concept-based models is convincing, it is somewhat concerning about the reliability of the formal concept lattices constructed for real-world datasets.
Figure A15 indicates that the concept lattices are actually somewhat noisy.
For example, in imagenet 100 examples, “agama, banded gecko and green lizard” are sharing the attribute of “insect”, which is not true, and in the CIFAR100 result, “chimpanzee” is grouped with fruits. Since the proposed method heavily relies on the induced concept lattice, it would be helpful to clarify why the proposed method could remain effective despite such noises, and to provide more analysis on how the quality of concept lattice affects the performance.

- In the example of generated formal concepts in Figure 2, the level #1 corresponds to the most specific cases such as <{dog}, {animal, vertebrate, domesticated, canine, mammal}>, which contradicts definition 3.1 (it states that i=1 being the most general top level). The figure should be modified.

- While several ablation studies on the proposed method are appreciated, an ablation study on each loss term such as
$l_{BCE}^{attr}, l_{BCE}^{group}, l_{CE}$ is missing. Although the authors provide the sensitivity analyses for $\alpha$ and $\beta$, the results indicate that the lowest values for $\alpha$ and $\beta$ showed the best performance.
This makes it unclear whether just eliminating each loss term (i.e., $\alpha=0$ or $\beta=0$) might show better performance. A direct ablation of each loss term would be helpful to clarify this confusion and make this work more comprehensive.

- In Figure 1 and A10,  it is said that “attributes are learned by a FoCA CBM”, but it seems somewhat misleading. As I understand, the attribute vocabulary itself is predefined by expert annotations or LLM (GPT-3) and FoCA CBM is trained to predict these attributes at different layers, rather than learning the attribute itself.

- (Typo) In L724, the table reference is broken.

---

> ### Author Rebuttal · Authors · 2026-03-31
>
> Thank you for your positive feedback. We are encouraged you found the paper well-organized, the work well-motivated and our experiments extensive.
>
> - **Noise in the formal concept lattice:** Attributes of classes are things that the class is associated with. We appreciate this concern. The perceived noise is actually semantically valid: *agamas*, *banded geckos*, and *green lizards* are all insectivores, so '*insects*' correctly appears as a shared attribute; similarly, '*fruit*' is a primary dietary attribute of *chimpanzees*, which is why they group with *fruit* classes. More broadly, noise in the lattice is generally a function of noise in the underlying class-attribute incidence relation— FCA itself introduces no additional noise in the construction process. Furthermore, our empirical results across three datasets with varying annotation quality (expert-annotated AwA2, LLM-annotated CIFAR100 and ImageNet100) consistently show strong performance, suggesting the method is robust to annotation imperfections.
> - **Ablation on each loss term:** Thank you for raising this point. The primary supervision for the model comes from $l_{CE}$, with $l^{attr}\_{BCE}$ and $l^{group}_{BCE}$ acting as auxiliary losses to nudge the model towards following structure and ordering. Since this is a multi-objective optimization problem, there is naturally a trade off between accuracy and structure. We have updated Tab A9 and A10 on Imagenet100 with the other metrics to give a complete picture.
>
>
>     | **Beta** | **Test Acc** | **CI** | **DBI** |
>     | --- | --- | --- | --- |
>     | 0.01 | **91.92** | 0.573 | 1.862 |
>     | 0.1 | 86.94 | 0.566 | 1.816 |
>     | 1 | 85.22 | **0.549** | **1.712** |
>
>     | **Alpha** | **Test Acc** | **CI** | **DBI** | **Avg Attr Acc** |
>     | --- | --- | --- | --- | --- |
>     | 0.01 | **91.92** | **0.573** | **1.862** | 88.48 |
>     | 0.1 | 87.73 | 0.6218 | 2.006 | 92.55 |
>     | 1 | 88.6 | 0.656 | 2.024 | **95.21** |
>
>     As we can see from the $\beta$ ablation table, giving more weight to the class grouping term leads to more coherent cluster groups (as indicated by the lower CI and DBI values) but at the cost of accuracy. Similarly, in the $\alpha$ ablation table, increasing the weight of the attribute loss leads to higher attribute accuracy, but degrades accuracy and clustering quality. This highlights the critical role that the auxiliary losses play in the learning. Crucially, setting either α=0 or β=0 leads to dramatic accuracy drops (30.07 and 42.91 respectively vs. 91.92 at α=β=0.01); CI and DBI are also poor due to the lack of proper supervision, confirming both loss terms are essential.
>
>     | **Alpha, Beta** | **Test Acc** | **CI** | **DBI** |
>     | --- | --- | --- | --- |
>     | 0, 0.01 | 30.07 | 0.675 | 2.257 |
>     | 0.01, 0 | 42.91 | 0.641 | 2.075 |
>     | 0, 0 | 29.24 | 0.6835 | 2.3902 |
>
>     Overall, these results suggest that the auxiliary loss terms are most effective when used as light regularizers, striking a balance between accuracy and semantic structure, which is consistent with standard multi-objective learning setups. We have added all these results to the Appendix.
>
> - **AwA2 Leakage:** AwA2 has dense and significantly redundant attribute annotations, where attributes capture overlapping semantic information. We attribute this to be the reason why the model performs well even when a small subset of the attributes are suppressed. We observe the impact of this in practice: constructing the formal concept lattice for AwA2 takes noticeably longer, indicating a more interconnected attribute space. To probe this further, we increased the number of attributes we switch off (to 3%) and observe that this does reduce performance (~8% drop in test acc), however the drop is only slightly larger than what we observe with the GPT4 hierarchy. Overall, this suggests that leakage isn’t solely due to annotation quality but is also significantly influenced by attribute density.
> - **Typos and Correction in Eqn 1:** Thank you for pointing these out. We have fixed the lattice level numbering in Fig 2 and have corrected Eq 1 (your observation is correct) to $A^{\uparrow} = \lbrace m \in M \mid \forall g \in A:\ \langle g, m \rangle \in I \rbrace$, $B^{\downarrow} = \lbrace g \in G \mid \forall m \in B:\ \langle g, m \rangle \in I \rbrace$.
> We have updated Fig 1 and Fig A10 to say “attributes predicted by” instead and have fixed the broken table reference.
>
> We are happy to answer any other questions/concerns you may have.

---

> > ### Author Rebuttal · Reviewer_8xYn · 2026-04-04
> >
> > I thank the authors for the additional experiments and clarifications, which addressed all of my concerns.
> > In particular, the ablation study on the $\alpha$ and $\beta$ and detailed explanation was informative.
> > Therefore, I decided to maintain my score.

---

> > > ### Author Response · Authors · 2026-04-08
> > >
> > > Thank you for your response. We are glad that your concerns are addressed and will incorporate these results and clarifications into the final version. If you feel that these additions further strengthen the paper, we would be grateful if you could consider reflecting this in your score.

---

### Official Review · Reviewer_fuEv · 2026-03-13

**Soundness:** 3
**Presentation:** 3
**Significance:** 2
**Originality:** 3
**Overall Recommendation:** 4
**Confidence:** 4

**Summary:**

Formal Concept Analysis (FCA) extracts a hierarchy of concepts from a learned neural network. The paper proposes to use concept hierarchies (derived from existing FCA method) for conditioning hierarchy aware training in Concept Bottleneck Models. They condition earlier layers in CBM on upper hierarchy levels of FCA (more abstract concepts)  and later layers on granular/detailed concepts.
The authors show improvements across CBM baselines in terms of accuracy as well as, concept segregation in terms of cluster impurity and cluster compactness in representations.

The authors also show that their method is not limited to FCA but can use any concept hierarchy satisfying some properties of formal concept lattices. The authors demonstrate successful intermediate layer interventions due to their layer wise concept conditioned training.

**Compliance With Llm Reviewing Policy:**

Affirmed.

**Final Justification:**

The authors addressed my concern of baseline comparison (confirmation of baselines being given the same flattened concepts and FoCA CBM-N result). The authors also correctly highlight the annotation requirement for practical utility. I agree with authors arguments in the rebuttal and hence update my rating.

**Key Questions For Authors:**

Please check SW1.

**Limitations:**

yes

**Strengths And Weaknesses:**

** Strengths**

* [S1] **Novelty**: The idea to introduce hierarchical concept supervision in CBMs is novel.
* [S2] **Intermediate layer interventions and interpretability**: due to explicit conditioning on the concepts, the intermediate layers are interpretable and interventions yield better results since the layers where class predictions started getting wrong can be identified (from class attributes for that layer).
* [S3] **Not limited to FCA**: The work is not limited to FCA but usable with other concept hierarchies (as shown in LLM experiment) which enhances its usability.
* [S4] **Decent Presentation to get the main idea**: The paper is generally well written with the key contributions clarified well.

**Weaknesses**

**Major Weaknesses**
* [MW1] **The baseline comparisons are not fair**: The authors use concept hierarchies which might have contributed to the improvements over baselines. For a fair baseline comparison, authors shall provide the labels to the baselines as well in some way.

* [MW2] **Real world Utility**: The neural networks are known to extract low level features in lower layers and high level in deeper layers; authors explicitly condition CBMs on such features/concepts which requires explicit hierarchical concept sets which will not be available in real world cases.

**Minor Weaknesses**
* [SW1] **Determining the layer for conditioning the concept level**: Exactly determining which layer shall be conditioned on which levels does not seem straightforward. Different layers usually learn such hierarchal concepts on their own: shallow layers are known to attend to more abstract/general features and deeper layers attend to specific features.
How do authors ensure they are selecting the correct layer?

While I acknowledge author's attempt to make each layer interpretable, I don't think the paper has a practical applicability [MW2] and seems to work because of over-supervision [MW1].

---

> ### Author Rebuttal · Authors · 2026-03-31
>
> Thank you for your thoughtful feedback. We are glad you found our work novel, our method general and the paper well-written. We address the raised questions and comments below.
>
> - **Baseline comparison not fair:** We respectfully disagree. All methods, including ours, **operate on the same concept and class sets**; the difference lies solely in how concepts are organized. Baselines treat concepts as a flat set at a single layer; FoCA CBMs structure them hierarchically via a formal concept lattice at intermediate network positions.
>
>     To address the concern that simply having more layers drives performance, we included the FoCA CBM-N (Naive) variant in Table 1, which places all lattice-derived attribute layers sequentially after the backbone and trains them like a standard CBM,  i.e., more layers, but no hierarchical intermediate supervision. FoCA CBM generally outperforms FoCA CBM-N, particularly on clustering metrics (CI: 0.573 vs 0.665, DBI: 1.862 vs 2.150 on ImageNet100), demonstrating that the gains come from *where* concepts are learned, not *how many* layers are used. Table 3 further confirms that FoCA CBMs incur the same FLOPs as Vanilla CBMs, ruling out computational overhead as a confound.
>
> - **Limited practical applicability:** We beg to differ, our method is relevant in any setting where (1) classes have associated attributes and (2) interpretability matters -- both conditions are common in real-world applications.
>
>     *High-stakes domains:* Concept-based models (see [CBM survey](https://www.researchgate.net/profile/Zhang-Kun-32/publication/399898851_Concept_Bottleneck_Models_for_Explainable_Decision_Making_A_Survey_of_Progress_Taxonomy_and_Future_Directions/links/696ef597c454e61a7f5b28c0/Concept-Bottleneck-Models-for-Explainable-Decision-Making-A-Survey-of-Progress-Taxonomy-and-Future-Directions.pdf) for its current breadth) are inherently interpretable by design; unlike post-hoc methods, they expose the reasoning process of the model through human-understandable concepts at prediction time, making them directly actionable for domain experts. This is valuable in high-stakes domains where model transparency is not optional but required. As stated in our Impact Statement, healthcare and medical diagnostics are a natural fit. A model diagnosing pathologies could learn general anatomical concepts (e.g., 'organ present', 'tissue type') in early layers and specific diagnostic markers (e.g., 'irregular border', 'calcification') in deeper layers, respecting the natural diagnostic hierarchy clinicians follow. Our multi-level intervention mechanism (Sec 6) is particularly valuable here, allowing practitioners to correct the model at the appropriate level of semantic granularity when required.
>
>     *Attribute availability:* Our method does not require a pre-specified or manually curated hierarchy. We construct our formal concept lattice from the same attribute and class annotations used in all baselines. In real-world settings where annotations are unavailable, we follow standard practice (as in works like [LFCBMs](https://arxiv.org/abs/2304.06129)) and obtain attributes via LLM queries, as we do for CIFAR100 and ImageNet100. The lattice is constructed as a one-time offline step taking 4.5s and 2s respectively for these datasets (Sec 6, Scaling and Practicality), making it practical even for new domains.
>
>     *Flexible Hierarchy:* Our framework is not restricted to FCA-derived hierarchies: Sec 6 demonstrates that an LLM-generated hierarchy (may not formally be an FCA lattice) satisfying subset-superset ordering works seamlessly as a drop-in replacement, further lowering the barrier to real-world deployment.
>
> - **Layer selection:** The layer selection is handled by our **class-cluster density** alignment strategy (Sec 3.2). For each block *j*, we apply *k*-means ($k=n$) to feature embeddings and measure $D_j$, the average distinct class labels per cluster. High $D_j$ indicates general  features; low $D_j$ indicates class-specific separation. We define an analogous quantity $\bar{D}_i$ for each lattice level *i* as the average extent size of formal concepts at that level, and assign lattice level $L_i$ to the earliest block *j* where $D_j \leq \bar{D}_i$.
>
>     This is visualized in Fig 3, which confirms the strategy recovers the expected coarse-to-fine structure without manual specification. Our strategy exploits the network's natural tendency to learn general-to-specific features by measuring it empirically, yielding a principled data-driven alignment. This is validated by consistent drops in accuracy and clustering metrics in Table A7 and A8 when layer-level assignments are varied.
>
>
> We are happy to answer any other questions/concerns you may have.

---

> > ### Author Rebuttal · Reviewer_fuEv · 2026-04-03
> >
> > I thank the authors for the detailed comments.
> >
> > Adding more layers does make the FoCA-CBM-N second best method in the ImageNet100 and best on CIFAR-100 in terms of accuracy and is a useful contribution given that it performs well even without utilizing additional information of concept hierarchies.
> >
> > I agree on its applicability in high stake domains; although explicit conditioning on additional concept hierarchies and training using additional layers makes the model more prone to errors due to issues in concept hierarchies (as correctly highlighted by authors in limitations) which can always be checked by domain experts in high stake environments.

---

> > > ### Author Response · Authors · 2026-04-08
> > >
> > > Thank you for your response and updated score. We would like to just clarify one point: a FoCA CBM and FoCA CBM-N use the same information from the lattice to obtain attribute supervision. The key distinction is that in a FoCA CBM-N, all the attribute layers are all learned after the backbone rather than at intermediate positions. Although this leads to competitive test accuracies, it struggles to capture structure (CI, DBI for a FoCA CBM-N are worse than a Vanilla CBM on Imagenet100). This highlights the importance of _where_ concepts are learned in the network. We have added a figure to the Appendix to illustrate this more clearly [[architectures](https://anonymous.4open.science/r/focacbm_results3-0371/cbm_architectures.pdf)].
> > > If you feel that these clarifications further strengthen the paper, we would be grateful if you could consider reflecting this in your score.

---

### Official Review · Reviewer_HuJW · 2026-03-13

**Soundness:** 3
**Presentation:** 2
**Significance:** 4
**Originality:** 4
**Overall Recommendation:** 5
**Confidence:** 4

**Summary:**

The paper introduce a generalization of Concept Bottleneck Models rooted in Formal Concept Analysis (FCA).
FCA can be naturally used to create a hierarchy of concepts by investigating the different layers of lattices that can be used to identify the right abstraction at different levels of the network. In particular, the lattice layers are associated each to a different layer of the network. The information provided by the lattice is then used to guide the training of the network, based on the assumption that hierarchical concepts can be useful to guide the network training and are likely be already extracted by the same network.

**Compliance With Llm Reviewing Policy:**

Affirmed.

**Key Questions For Authors:**

- Q1: Is the approach also potentially usable post-hoc to detect whether existing neural network have learnt a hierarchy of concepts?

**Limitations:**

Yes

**Strengths And Weaknesses:**

## Strenghts:
The paper is very interesting, proposing a quite natural but very well-grounded approach for guiding the network training and extedining the CBM paradigm to multiple layers in the network.
- **Soundness:**, the paper is technically sound, the theoretical framework is very solid, equipped with theoretical analysis and a few strong experimental results.
- **Significance**: the paper propose a novel, theoretically grounded approach obtaining very good results in a hot explainable ai topic (CBM): it is extremely likely that it will have a big impact
- **Originality**: the proposed approach is quite novel, although hierarchical cbms have been investigated previously, the idea of applying multiple different concept layers at different places in the network it


## Weaknesses:
- **Soundness**: although the paper is overall sound, the experiments other than Table 1 are not scientifically conducted.
  - The intervention  Multi-Level Interventions experiment is reported in Fig. 5 b on a single dataset, with a single compared baseline represented by intervention on the same architecture at different layer. I strongly think that this experiment should have been extended to most of the compared baselines and also it should have followed the standard practice for this experiment, where number of oracle intervention are reported w.r.t. accuracy gains.
  - The results on the clustering are very interesting but are not reported for in any other form if not the metrics, some qualitative examples would have been interesting to see (after many theoretical diagrams).
  - Finally the results on the ViT backbone are reported for CBM and Foca only. Reporting them for all baselines (or at least a more cospicuos subset) would have been better. It would have been particularly important since one of the strongest claim in the paper is that neural network naturally create hierarchical representation in the network layers: while this has been deeply validated in CNNs, in ViT and transformers in general the question is still open, with several methods showing that representations in ViTs are quite uniform across layers, as self-attention aggregates global context right from the first layers, and don't necessarily build local-to-global hierarchies [1], [2].
- **Presentation**: The presentation is where the paper is mostly lacking. I understand that the theoretical framework to present is important and there are many concepts to present, but the presentation is very confusing I got lost many times when reading the Method section. The class-cluster density appears out of nowhere. The way in which the output of each layer s_j is computed is very unclear and entirely not justified. Also, it is possible that I missed some results in the appendix, but it is quite difficult to navigate 21 pages of appendix without having the precise reference to look at in the main paper (it is always referred as the Appendix without precise references). Finally also the notation is quite lacking, the following is a non-exhaustive list of notation issues:
   - Eq.1, it is introduced as "Given A ⊆ G, B ⊆ M, this is defined as:" but 3 equations appear. Also what "this" was is not clear
   - Eq.1, the set are badly defined with two "such that" symbols employed (: and | ). I guess the second should have been a comma?
   - Eq.1, the constraint for the variables constituting set are defined in terms of other variables (x,y). i guess it should have been (m,g)?
   - After Eq.2 fc.intent is correctly introduced, but it is not clear why also fc.extent is introduced.
    - Line 232, right column. The indicator function should be indicated with \mathbb. and since the condition is at subscript, brackets are not necessary.
   - Eq.5 the fourth log is in italic and it is not clear why.

As a suggestion, I would dedicate more space to the presentation of the method (possibly including a roadmap at the beginning of section 3) and of the results (including more relevant results in the main paper), while reducing the visualization diagrams and possibly also the theoretical results (which take more than one page alone)

---

> ### Author Rebuttal · Authors · 2026-03-31
>
> Thank you for your positive feedback. We are encouraged that you found our paper very interesting, novel, technically sound and theoretically grounded. We address your comments below.
>
> - **Multi-level interventions:** Our multi-level intervention strategy is based on misclassification severity, which is why Fig 5b focuses exclusively on the misclassification set — as misclassification sets differ between models, the ratio of corrections is model-specific and cross-model comparison would be unfair.
> Nonetheless, we agree with the suggestion and have extended this to the whole test set, following the standard evaluation protocol (as in CEMs, ProbCBMs), and performing k random oracle interventions on general and specific attribute layers compared against 3 baselines. We plot # oracle interventions vs accuracy gains in [[interventions plot](https://anonymous.4open.science/r/focacbm_results-43BF/inet100_wholetestset_interventions.pdf)] and find that our general-layer interventions yield the highest accuracy gains per oracle intervention. We report this as an additional study in the Appendix.
> - **Clustering qualitative results:** Thank you for this suggestion. We have examined  clusters from the blocks of a FoCA CBM and a Vanilla CBM. Sample results are at [[qualitative results](https://anonymous.4open.science/r/focacbm_results-43BF/clusters_qualitative_results.pdf)]. We see that classes from clusters of a Vanilla CBM are mixed up across blocks (dense earlier, and suddenly transition later), whereas a FoCA CBM gradually telescopes relevant concepts. For example, in the result on the top with *black swans*, the classes in a Vanilla CBM’s clusters only separate out into the group {*black swan, goose*} at the last block, being quite noisy throughout. However in a FoCA CBM, we observe gradual semantic refinement. The first block captures a mixture of classes, the second block refines it to classes that are associated with water, the third block narrows this down to a set of birds and finally the last block has a separate cluster for {*black swan*}. We have added these results to the Appendix.
> - **ViT Results:** We obtained results on more baselines with ViT backbones on CIFAR100 (results below); our method continues to outperform baselines.
>
>
>     | **ViT-Based Model** | **Test Acc** | **CI** | **DBI** |
>     | --- | --- | --- | --- |
>     | ViT CBM | 84.49 ± 0.55 | 0.774 ± 0.016 | 2.237 ± 0.006 |
>     | LFCBM | 72.40 ± 0.27 | 0.873 ± 0.000 | 2.797 ± 0.000 |
>     | Posthoc CBM | 71.29 ± 0.63 | 0.873 ± 0.000 | 2.797 ± 0.000 |
>     | LaBo | 76.12 ± 1.04 | 0.873 ± 0.000 | 2.797 ± 0.000 |
>     | CFCBM | 74.11 ± 0.44 | 0.873 ± 0.000 | 2.797 ± 0.000 |
>     | FoCA ViT | **86.65 ± 0.30** | **0.755 ± 0.004** | **1.983 ± 0.021** |
>
>     Regarding representations of ViTs, we agree that their nature is less explicitly understood than CNNs. One could view our method as providing an explicit inductive bias for hierarchical representations on such architectures. Recent works like [Park et al. (ICLR’ 25)](https://openreview.net/forum?id=bVTM2QKYuA) show that hierarchical structure can emerge in the representation space of transformers. Our results provide empirical evidence that FoCA's inductive bias successfully uncovers this structure in ViTs, as reflected by consistent gains across all three metrics.
>
> - **Post hoc usefulness:** This is an interesting direction. Although our method is primarily intended to induce explicit hierarchical semantics during training, it could in principle be applied post-hoc. Given a concept set and a trained network, our class-cluster density measure could identify which layers are most semantically aligned with which lattice levels. One could then probe those layers for the corresponding concepts and measure ordering violations, i.e., how often a specific concept activates in a layer where its parent concept does not, to quantify the degree to which the network has implicitly learned a concept hierarchy. We see this as an interesting direction for future work.
> - **Presentation:** Thank you for pointing these out. We have updated Fig 2 ([updated main figure](https://anonymous.4open.science/r/focacbm_results-43BF/updated_main_figure.pdf)) to include class-cluster density and have added a roadmap at the beginning of the method section. $s_j$ is the classifier output at position j in the network (stated in L247rt). We have now clarified its computation: it is the output of classifier $p_j$ applied to attribute activations obtained from $q_j$, which takes the global average pooled feature $f_j(x)$ as input. We have corrected Eq 1; ”this” is the condition attribute-object subsets have to satisfy in order to be formal concepts. We have corrected all other typos and have updated all Appendix references to point to specific Appendix sections.
>
> We are happy to answer any other questions/concerns you may have.

---

> > ### Author Rebuttal · Reviewer_HuJW · 2026-04-01
> >
> > I thank the author for their answers, and I am happy that some of my suggestions have been helpful.
> > I still think that at least one qualitative result must be included in the main paper to improve the presentation and its comprehension further.
> > Anyway, I will raise my score to accept.

---

> > > ### Author Response · Authors · 2026-04-08
> > >
> > > Thank you for your response and updated score. We agree that including a qualitative result in the main paper would improve clarity, and will incorporate a representative example in the final version, along with the other clarifications discussed during the rebuttal.

---

### Official Review · Reviewer_4yz5 · 2026-03-18

**Soundness:** 3
**Presentation:** 3
**Significance:** 3
**Originality:** 2
**Overall Recommendation:** 4
**Confidence:** 4

**Summary:**

This paper is about interpretable-by-design models, and, more precisely, it proposes an extension of the well-known concept-based bottleneck models. The motivation of this paper is to integrate the semantic structure of concepts (represented here by a concept lattice in formal concept analysis) into the concept learning principle. The concept lattice obtained by formal concept analysis imposes an order relation on the concepts, and, more precisely, formalizes the generalization/specialization relationship. This ordering is used to define both attribute and class labels for different blocks of layers in the network, rather than a single concept layer as in the original concept bottleneck. The paper also includes a theoretical analysis of hierarchical consistency. An experimental validation is performed on three datasets: ImageNet100, CIFAR100 with LLM-generated attributes, and AwA2 with expert-annotated attributes. An important ablation study is also provided in the supplementary material.

**Compliance With Llm Reviewing Policy:**

Affirmed.

**Final Justification:**

The authors’ responses have convinced me of the paper’s strength and its relevance to the field.

**Key Questions For Authors:**

+ The class-cluster density is based on a simple k-means with $k=n$. Choosing the k-means algorithm with k set to the exact number of classes assumes a strong hypothesis about the data distribution. How to justify the choice?

+ The definition of $\mathcal{Y}(K)$ is not clear for me. Does it correspond to the number of different ground-truth class labels or just the set of ground-truth class labels? In the second case, is it just the number of samples in K? It also makes the definition of equation 3 ambiguous.

+ As the concept lattice can be seen as formal knowledge, how to integrate recent works on [semantic loss](https://arxiv.org/abs/1711.11157) or a [semantic probabilistic layer](https://arxiv.org/abs/2206.00426) into the approach?

+ In the original CBM model, various learning schemes are proposed: independent, sequential, and joint. What about independent and sequential schemes for the proposed FoCA CBM approach?

**Limitations:**

yes

**Strengths And Weaknesses:**

**Main Strengths**
+ The problem addressed in the paper is significant and interesting. The lack of semantic structure in the concept-bottleneck problems limits their usage.
+ The paper is well-written and structured. The provided illustrations help clarify the paper. The paper is also well balanced between methodology, theoretical analysis, and experimental studies.  The supplementary material shows the depth and seriousness of the experimental sections.
+ The proposed approach is generic, and other semantic structures satisfying the subset/superset order can be used.

**Main Weaknesses**
+ The paper lacks a clear positioning and lacks related work. For instance, in the concept-bottleneck model family, how does the proposed approach compare to [Barbiero et al., 24](https://arxiv.org/pdf/2308.11991v2) or [De Felice et al., 25](https://openreview.net/forum?id=UX143QGvb8)?
+ CBMs approaches are supervised concept-based approaches. For many reasons, a big part of the community studies unsupervised concept-based models, also with the semantic structure constraints.
+ Some methodological choices could be more motivated and positioned with alternative approaches. See for questions sessions.
+ The alignment strategies between the neural network blocks and the layers of the lattice are not sufficiently motivated and are highly dependent on the model. The notion of 'block' could also be more defined.
+ The originality of the paper is not sufficient for the ICML conference. The paper introduces a new interesting methodological approach, merging CBM and FCA domains, but it is an incremental contribution.

---

> ### Author Rebuttal · Authors · 2026-03-31
>
> Thank you for your thoughtful feedback. We are glad you found the problem significant, the paper well-written and the work well-balanced. We address your concerns below.
>
> - **Positioning & related work:** RCBMs (Barbiero et al 2024) model relations between concepts via GNNs, and C²BMs (De Felice et al 2025) introduce a causal bottleneck — both incorporate structure over concepts but retain a single concept layer before the final classifier. FoCA CBMs differ fundamentally by distributing concept learning across network depth, guided by the order-theoretic properties of a formal concept lattice (Sec 4). In short, RCBMs and C²BMs structure the concept space; we structure the representation space hierarchically. (Added this to related work.)
> - **Why CBMs over unsupervised concept-models:** While unsupervised methods (e.g., SENN, CLIP-Dissect, ProtoVAE; note TCAV and ACE are post-hoc) have merit, supervised concept learning remains widely studied (see [CBM survey](https://www.researchgate.net/profile/Zhang-Kun-32/publication/399898851_Concept_Bottleneck_Models_for_Explainable_Decision_Making_A_Survey_of_Progress_Taxonomy_and_Future_Directions/links/696ef597c454e61a7f5b28c0/Concept-Bottleneck-Models-for-Explainable-Decision-Making-A-Survey-of-Progress-Taxonomy-and-Future-Directions.pdf) for its current breadth). It provides explicit semantic interpretability critical in high-stakes domains, enables principled formal hierarchies via subset-superset relations, and supports test-time interventions — all not well-defined for unsupervised concepts. LLM-based annotation has also significantly reduced labeling cost (used for 2 of our datasets). Our hierarchical learning contribution is orthogonal to the supervision source and could guide unsupervised concept discovery as well. We see these as complementary directions.
> - **Incorporating semantic loss:** The parent-child relations in the lattice (e.g., feline → mammal) naturally lend themselves to semantic constraints. We implemented a weighted attribute loss on ImageNet100 where specific attribute probabilities are conditioned on general ones, penalizing cases where a general attribute has low probability but a specific one has high probability e.g. P(mammal)=0.2, P(feline)=0.8. We observe faster convergence in early epochs with comparable final accuracy (~91%). Our iterative class-group refinement already achieves something analogous for classes, which may explain the equivalent accuracies. We see this as interesting future work.
> - **Defining a block:** We have added to Sec 3:  *'Modern neural network architectures are organized into progressive modular components, for example, residual blocks in ResNets, transformer encoder blocks in ViTs, etc. We refer to these generically as `blocks' and treat them as candidates for semantic supervision.'*
> - **Definition of $\mathcal{Y}(K)$:** $\mathcal{Y}(K)$ is the **set** of distinct ground-truth class labels of samples in cluster $K$ (see L199rt). For example, if $K=\{(x1,y1),(x2,y2),(x3,y2)\}$ , then $\mathcal{Y}(K) = \{y_1, y_2\}$ and  $|\mathcal{Y}(K)|=2$. Eqn 3  averages this count across all $n$ clusters. We have clarified this in the text.
> - **Motivation for class cluster density, why k=n:** Our goal of class-cluster density is not to model the data distribution but to measure discriminative capacity at each block, analogous to linear probes for representation quality. Setting *k=n* provides a natural, consistent reference point aligned with the classification task. Any reasonable proxy for representational granularity would serve this purpose; *k=n* is a principled default here, given the setting.
> - **Independent and sequential FoCA CBMs:** We trained FoCA CBMs on ImageNet100 under all three schemes and observe a significant performance gap (Joint: 91.88±0.35, Sequential: 86.69±0.04, Independent: 77.54±0.35). We attribute this to two factors: (1) Broken iterative refinement: training attribute layers without class feedback breaks the coordination our cascading mechanism relies on; (2) Compounding error: errors at earlier layers cascade through the hierarchy. Joint optimization is thus essential, and high attribute accuracy in isolation is insufficient. We have added this analysis to the Appendix.
> - **Originality:** We respectfully disagree. We are the first to bring FCA into concept-based learning in vision, a non-trivial connection acknowledged by other reviewers, and the first to distribute concept learning across network depth. We additionally introduce class-cluster density-based alignment, iterative class-group refinement, and severity-based multi-level interventions, with theoretical grounding showing our supervision minimizes ordering violations and yields parsimonious attribute sets. We believe this goes well beyond an incremental contribution.
>
> We are happy to answer any other questions/concerns you may have.

---

> > ### Author Rebuttal · Reviewer_4yz5 · 2026-04-03
> >
> > I thank the author for their answers and for addressing all my concerns.
> >
> > Although I remain convinced that a more in-depth analysis and greater theoretical grounding—for example, by drawing more heavily on geometry—regarding the choice and contribution of FCA to a better structuring of the representation space could significantly enhance the paper’s value, the authors’ responses have convinced me of the paper’s strength and its relevance to the field. I would also like to revisit my comment on originality and agree with the authors that I need to reconsider my assessment.
> > The different answers and clarifications have to be added in the revision.
> >
> > I have adjusted my score accordingly.

---

> > > ### Author Response · Authors · 2026-04-08
> > >
> > > Thank you for your response and updated score. We are especially grateful for your positive reassessment of the paper’s originality and overall contribution. We also agree with the suggestion regarding providing a geometric perspective on the impact of FCA on the representation space. To this end, we would like to highlight an interpretation based on the **maximal rectangle** characterization (Ganter & Wille, 2024) from the FCA literature.
> > >
> > > A formal concept $\langle A, B \rangle$ is precisely a maximal rectangle or biclique in the incidence matrix $I$: $A$ is the maximal set of objects sharing all the attributes in $B$ and $B$ is the maximal set of attributes sharing all the objects in $A$ (Eq 1). This maximality has a direct geometric implication in the representation space. Because each concept encoder $q_j$ is trained to learn a maximal attribute set for each corresponding class group, the representation space at layer $j$, is encouraged to get partitioned into regions corresponding to class groups that are as tight as possible. Across depth, due to our iterative refinement, this induces progressively finer partitions whose boundaries are drawn out by the addition of more specific attributes.
> > >
> > > This is geometrically much stronger than what an LLM-hierarchy can guarantee. An LLM-hierarchy may not produce attribute sets that are maximal, implying that some discriminative directions are either redundant or missing, which is precisely what we observe empirically - our GPT4-based hierarchy shows significant concept leakage (Tab 2). To further formalize this, we introduce the notion of a *closure margin ($\mu$),* which intuitively measures the thickness of the “buffer zone” that separates a formal concept from the rest of the space. FoCA CBMs have $\mu \geq 1$ implying a non-zero separation, i.e. every class outside $A$ disagrees with at least one attribute in $B$. This directly quantifies concept leakage resistance - the minimum number of attribute flips needed for an outsider class to enter the concept. Any non-closed (e.g. LLM-derived) block has $\mu = 0$ and is *already leaked* in the noiseless incidence matrix. This is a theoretically backed explanation for the concept leakage gap in Table 2 and the tighter cluster structure of FoCA CBMs.
> > >
> > > We hope this, combined with the other theoretical results (Theorems 4.1 and 4.2) and the empirical evidence, further strengthens the case for the principled role of FCA in our framework. We will incorporate this perspective in the Appendix, along with the other clarifications discussed in the rebuttal. If you feel that these clarifications further strengthen the paper, we would be grateful if you could consider reflecting this in your score.

---

### Decision · Program_Chairs · 2026-04-30

**Decision:**

Accept (regular)

**Comment:**

The paper relies on Formal Concept Analysis (FCA) to extend Concept Bottleneck Models. The FCA allows to build a lattice to represent coarse to fine-grained concepts, instead of relying on a single bottleneck layer. These networks are "interpretable by design".

Overall, reviewers found the paper clear and well-written. The idea is praised for its originality, and its broad applicability. Empirical results were found to be convincing.

Some concerns were raised about unfair comparison to baselines, which were addressed during rebuttal.

Given the relevance of the approach to the field, I recommend acceptance.